# From Flatland to Space: Teaching Vision-Language Models to Perceive and Reason in 3D

Jiahui Zhang[1][*]    Yurui Chen[1][*]    Yanpeng Zhou[2,*]    Yueming Xu[1]    Ze Huang[1]
Jilin Mei[1]    Junhui Chen[1]    Yu-Jie Yuan[2]    Xinyue Cai[2]    Guowei Huang[2]
Xingyue Quan[2]    Hang Xu[2]    Li Zhang[1]†

[1] School of Data Science, Fudan University    [2] Huawei Noah's Ark Lab

https://LogosRoboticsGroup.github.io/SPAR

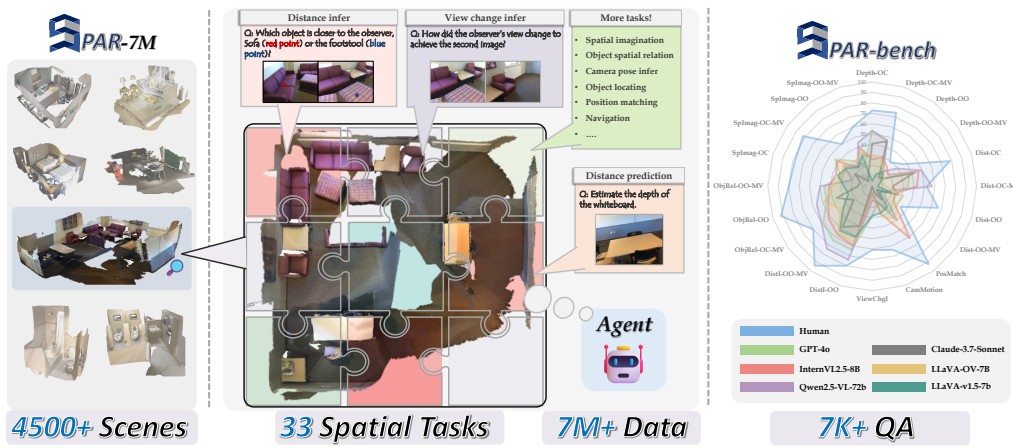

Figure 1: Overview of our *Spatial Perception And Reasoning (SPAR)* dataset and benchmark. Our dataset is sourced from 4,500 scenes and comprises 33 spatial tasks spanning single-view, multi-view, and video settings. Our benchmark includes over 7,000 carefully curated high-quality samples to comprehensively evaluate the spatial understanding capabilities of existing models.

## Abstract

Recent advances in LVLMs have improved vision-language understanding, but they still struggle with spatial perception, limiting their ability to reason about complex 3D scenes. Instead of injecting 3D representations, we unlock VLMs using spatially relevant 2D images. To this end, we introduce a novel 2D spatial data generation and annotation pipeline built upon scene data with 3D ground-truth. This pipeline enables the creation of a diverse set of spatial tasks, ranging from basic perception tasks to more complex reasoning tasks. Leveraging this pipeline, we construct *SPAR-7M*, a large-scale dataset generated from thousands of scenes across multiple public datasets. In addition, we introduce *SPAR-Bench*, a benchmark designed to offer a more comprehensive evaluation of spatial capabilities compared to existing spatial benchmarks, supporting both single-view and multi-view inputs. Training on both *SPAR-7M* and large-scale 2D datasets enables our models to achieve state-of-the-art performance on 2D spatial benchmarks. Further fine-tuning on 3D task-specific datasets yields competitive results, underscoring the effectiveness of our dataset in enhancing spatial reasoning.

[*]Equal contribution. †Corresponding author (lizhangfd@fudan.edu.cn).

39th Conference on Neural Information Processing Systems (NeurIPS 2025) Track on Datasets and Benchmarks.

| Dataset | Perception | Reasoning | Single view | Multi-view[†] | Video | Task types | QA quantity |
|---|---|---|---|---|---|---|---|
| ScanQA [13] | ✗ | ✓ | ✗ | ✗ | ✓ | - | 27k |
| SQA3D [14] | ✗ | ✓ | ✗ | ✗ | ✓ | 16 | 33k |
| 3D-LLM [8] | ✗ | ✓ | ✗ | ✗ | ✓ | 9 | 659k |
| LEO [15] | ✗ | ✓ | ✓ | ✗ | ✓ | 9 | 513k |
| SceneVerse [16] | ✗ | ✓ | ✗ | ✗ | ✓ | 3 | 2.5M |
| MMScan [17] | ✓ | ✓ | ✗ | ✗ | ✓ | 20 | 6.9M |
| *SPAR-7M* | ✓ | ✓ | ✓ | ✓ | ✓ | **33** | **7M** |

Table 1: **Spatial understanding datasets comparison.** [†] Multi-view uses 3–5 images per QA pair.

# 1 Introduction

Recent advances in Large Vision-Language Models (LVLMs) have significantly enhanced vision-language understanding [1, 2, 3, 4, 5]. However, to empower VLMs to better interact with the physical world, spatial understanding is essential, serving as a cornerstone of embodied intelligence. Nevertheless, current VLMs exhibit limited spatial perception, often struggling with comprehensive understanding and reasoning about spatial relationships in complex scenes [6, 7, 8, 2, 9, 10].

A recent line of research explores enhancing spatial understanding by incorporating 3D representations of scenes [8, 9, 10, 11, 12]. However, these methods encounter several challenges. First, 3D data are scarce, with suboptimal quality and uneven distribution, making them unsuitable for large-scale pretraining. Second, designing a model (e.g., a point cloud encoder) to effectively capture 3D information is inherently challenging, particularly since complete 3D representations are often inaccessible in real-world scenarios. Moreover, these approaches modify the network architecture, hindering seamless integration with existing VLM frameworks [5, 3].

This raises the question: Is it truly necessary to incorporate 3D representations into VLMs? For humans, spatial understanding of the relationship between surrounding objects is innate, relying on the ability to observe and implicitly reconstruct the entire space from 2D observations. Motivated by this capability, we seek to empower VLMs to acquire spatial understanding directly from large-scale 2D data (e.g., image inputs), enabling them to perform spatial reasoning without explicit 3D supervision. To facilitate this, we construct a spatially relevant dataset tailored for training and evaluation. Additionally, to enable VLMs to tackle a broader range of 3D tasks, we design a multi-view-based 3D grounding approach.

We categorize spatial understanding into two levels: **spatial perception**, which encompasses fundamental spatial concepts such as depth, distance, relative positions, and camera viewpoints, representing low-level cognitive abilities; and **spatial reasoning**, which involves higher-order spatial understanding, requiring the ability to infer and construct 3D scenes implicitly.

Building on these two levels of spatial understanding, we introduce *SPAR-7M*, a dataset comprising 33 tasks that span a broad range of complexities. To facilitate systematic learning and evaluation, we organize all tasks into three levels—low, medium, and high—reflecting a natural progression from spatial perception to spatial reasoning. While previous datasets have predominantly focused on semantic descriptions, *SPAR-7M* places equal emphasis on **numerical** representations, enabling models to develop a more accurate and grounded understanding of 3D space. For example, we include fundamental perception tasks such as depth and distance estimation. We argue that under multi-view inputs, pixel- or feature-level matching is a foundational capability that supports higher-level spatial reasoning. Accordingly, many of our tasks—such as cross-view object matching and viewpoint transformation prediction—are explicitly designed to evaluate and strengthen this essential skill.

To generate the necessary training data, we introduce a 2D data generation and annotation pipeline that derives rich supervision from 3D scene data. The resulting dataset, *SPAR-7M*, provides precise 3D ground-truth annotations, significantly improving both task diversity and reliability. Unlike existing approaches that rely solely on 2D annotations [6, 7], our dataset enables a more structured and accurate assessment of spatial understanding. Tab. 1 illustrates the diversity of input formats and task types within our dataset.

Despite the growing body of spatial benchmarks, many focus predominantly on high-level spatial reasoning, often overlooking fundamental spatial perception [18, 19, 17]. Furthermore, some benchmarks, particularly those using internet-collected images, struggle to effectively evaluate multi-view tasks, while others based on video inputs are not well-aligned with typical model inputs [20]. To

address these limitations, we select 20 tasks from the *SPAR-7M* validation set. After carefully manual validation and filtering, we created *SPAR-Bench*, a comprehensive benchmark consisting of 7,207 questions that span tasks from basic perception to complex reasoning. *SPAR-Bench* supports both single- and multi-view inputs, offering a more holistic and diverse evaluation of spatial understanding.

Our main contributions are as follows: **(i).** We present a 2D spatial QA generation pipeline using 3D ground truth to create both single- and multi-view data. This yields tasks covering numerical and semantic reasoning, forming a more comprehensive spatial understanding dataset. **(ii).** We present *SPAR-Bench*, a benchmark for evaluating the spatial capabilities of existing models. **(iii).** By training models on both *SPAR-7M* and large-scale 2D internet datasets, we achieve state-of-the-art performance across multiple 2D spatial benchmarks. Moreover, even with 2D-only inputs, further fine-tuning on 3D task-specific datasets yields competitive results.

## 2   Related works

**Datasets for spatial understanding.**   Existing multimodal datasets are mostly single-view or limited multi-view and aimed at retrieval/Basic VQA, making them insufficient for advanced 3D reasoning [21, 22, 23, 24]. Some datasets attempt to address this by collecting large-scale images from the internet, but they lack precise 3D annotations, relying instead on estimation models, which limits their effectiveness in multi-view and spatial reasoning tasks [6, 7]. Others incorporate image sequences or 3D scene representations for object counting, spatial relationship reasoning, or grounding, but they mainly focus on high-level semantics while neglecting low-level geometric cues such as depth, distance, and camera parameters [13, 14, 25]. Concurrently, datasets like MSR3D [26] are also emerging, offering large-scale, multi-modal inputs that explicitly include point clouds alongside text and images. These datasets are designed to evaluate situated reasoning and navigation within 3D environments, providing valuable benchmarks for models that directly process 3D data. Recent efforts [27] explore spatial reasoning from a robotics perspective, emphasizing reference-frame understanding and grounded spatial relations in real-world scenes. While these 3D-centric approaches (e.g., MSR3D) are invaluable for embodied AI and situated interaction, our work takes a different stance. We aim to achieve a broader spectrum of general 3D spatial tasks and a cognitive stratification of reasoning abilities, learned primarily from 2D visual data derived from 3D ground truth. This explores the extent to which VLMs can develop rich 3D understanding without explicit 3D input modalities during inference. To overcome these limitations and to further explore this potential, we propose a 3D-ground-truth-based data generation pipeline capable of rendering single-view, multi-view, and video-based data. Our approach ensures precise spatial supervision, covering a broad spectrum of tasks from basic perception to advanced reasoning, structured into three difficulty levels. This provides a more comprehensive and scalable solution that complements approaches leveraging direct 3D representations or focusing on specific robotic interactions.

**Benchmarks for evaluating spatial understanding.**   Existing spatial understanding benchmarks mainly focus on high-level reasoning while often neglecting fundamental spatial perception tasks [19, 18, 17, 13, 14, 20]. Most of them rely on either single-view images or video inputs, with limited consideration of multi-view settings, despite their ability to better represent local 3D structures. Single-view images restrict the complexity of the questions and thus provide an incomplete assessment of VLMs' spatial reasoning capabilities. Some works focus on multi-image spatial reasoning or general multi-image QA [28, 29, 30], while others rely on video inputs [20]; however, these settings either emphasize reasoning-centric/generic QA or require video-specific processing that many VLMs are not optimized for, and thus do not by themselves reflect the full spectrum of spatial skills. To address these limitations, we introduce *SPAR-Bench*, a more comprehensive benchmark that encompasses both fundamental and advanced spatial tasks. It supports both single- and multi-view inputs, enabling a more systematic and reliable evaluation of spatial understanding.

## 3   Method

Our research aims to equip VLMs with robust and accurate spatial understanding, focusing on both fundamental 3D perception and higher-order spatial reasoning and imagination. To address the challenges posed by limited data, we propose a novel data generation pipeline that creates a diverse range of spatial understanding tasks. Our pipeline leverages 3D scenes from ScanNet [31],

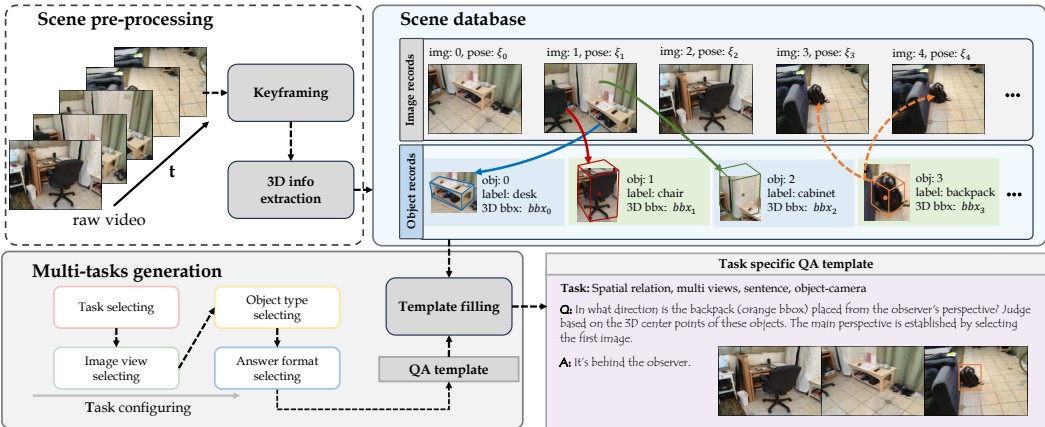

Figure 2: **Data generation pipeline.** Our pipeline consists of three main parts: (1) Filtering redundant images by sampling approach. (2) Extracting scene 3D information and creating a unified data structure to support multi-view cross-retrieval and annotation. (3) Leveraging metadata to automatically generate a variety of tasks and forms of QA data.

ScanNet++ [32], and Structured3D [33], ensuring a broad and diverse representation of indoor environments. Along with this, we introduce a new benchmark to provide a comprehensive evaluation of models' spatial capabilities. In the following sections, we first detail our data generation pipeline (Sec. 3.1) and the key characteristics of *SPAR-7M* (Sec. 3.2). We then introduce *SPAR-Bench* (Sec. 3.3), followed by an analysis of data quality and the manual filtering process used to ensure benchmark reliability (Sec. 3.4). Additionally, in Sec. 3.5, we present a multi-view-based 3D grounding approach designed to enhance VLMs' ability to tackle a broader range of 3D tasks.

## 3.1 3D-driven data generation pipeline

This section presents our data generation pipeline (detailed in Fig. 2), which takes as input a scene mesh, 3D object bounding boxes, camera parameters, and image sequences. Through filtering, 3D information extraction, and task construction, we produce diverse spatial understanding data.

**Filtering and subsampling.** In the first step, we focus on reducing data redundancy by filtering out images with minimal camera movement. Specifically, each image is associated with a camera pose. If an image's camera position is within a distance threshold of a retained image and its orientation differs by less than an angular threshold, it is considered redundant and removed. The whole process is similar to keyframing and the implementation details can be found in the Appx. 7.3.

**Extracting 3D information.** In the second step, we extract global 3D information from each scene and organize it into a unified data structure for cross-view retrieval and downstream annotation. This structure includes two types of entities: image records and object records, which together create a consistent mapping between image and object data in world coordinates. An image record stores image-level metadata, including index, camera parameters, resolution, and visible objects. Each object is linked to its 3D bounding box, 3D center in world coordinates, and 2D projection. An object record contains object-level metadata, including ID, category, and a list of associated image indices. This compact representation facilitates multi-dataset alignment and supports scalable task generation. Technical details are provided in Appx. 7.4.

**Multi-tasks generation.** In the third step, we use the complete 3D scene information to automatically generate tasks and corresponding QA pairs. Each task is configured with key details such as task type (e.g., distance prediction, spatial relations), input image count, question format, and the object types involved (object-to-object or object-to-observer relations). For each task, we follow a specific pipeline to generate annotations, ensuring flexibility to extend or customize tasks as needed. After generating the annotations, placeholders in the template (e.g., [label_A], [label_B], [left_right]) are replaced with actual object labels and computed spatial information. The templates are initially created by humans, expanded with the help of GPT, and then refined through manual selection to

ensure diversity and accuracy. Detailed information about task pipelines, QA templates, and GPT prompting can be found in the Appx. 7.5.

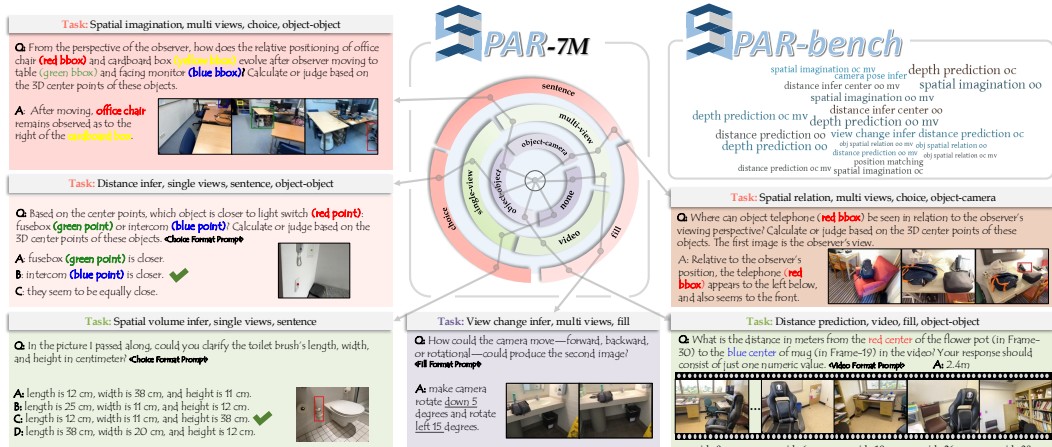

Figure 3: **Dataset and benchmark visualization.** We present illustrative samples from *SPAR-7M* and *SPAR-Bench* to highlight the diversity of spatial tasks and input types. The examples have been simplified for clarity; Full task formats are provided in the released dataset and Appx. 7.6.

## 3.2 Spatial dataset construction

Based on our data generation pipeline, we construct *SPAR-7M*, a large-scale dataset designed for spatial understanding. *SPAR-7M* consists of 4,000+ indoor 3D scenes from diverse sources, covering 33 task types with over 7 million QA pairs. Each QA pair is paired with camera parameters, and depth, enabling both multi-view reasoning and 3D spatial inference. The dataset includes tasks ranging from basic spatial perception to high-level spatial reasoning (detailed in Fig. 4).

We evaluate spatial understanding with three input formats—single-view, multi-view, and video. Single-view tasks assess spatial inference from one image, while multi-view tasks require cross-view integration. Video inputs introduce scene-level reasoning and long-range spatial perception. This diversity enables a comprehensive evaluation across settings. Tasks are further categorized by spatial relation type: object-object (OO) and object-camera (OC). OO tasks focus on inter-object positioning (e.g., left/right/behind), while OC tasks involve viewpoint-dependent relationships. For multi-view inputs, one image is randomly chosen as the primary viewpoint for reasoning.

## 3.3 Spatial benchmark construction

*SPAR-Bench* evaluates models across spatial tasks from low-level geometry to high-level relations. It includes 20 tasks sampled from the *SPAR-7M* test set, each with 400 QA pairs and manually verified for quality, resulting in a total of 7207 QA samples. The tasks span both single-view and multi-view settings and are formulated as either numerical prediction or classification, evaluated using mean relative absolute error (MRA) or accuracy, respectively. While the full dataset includes 33 spatial tasks, we select 20 for evaluation. Tasks requiring video input are excluded to disentangle spatial understanding from temporal modeling. Our focus is on evaluating how well models reason about 3D structure, geometry, and viewpoint relations

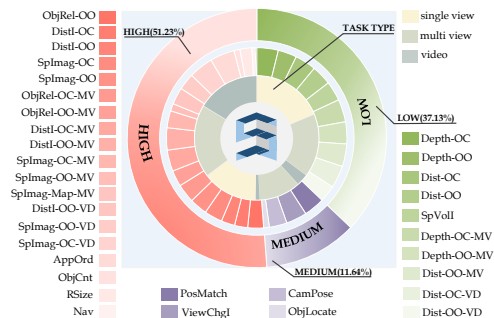

Figure 4: **Dataset statistics.** Our Dataset can be categorized into high, medium, and low three levels, totaling 33 subtasks.

from one or more views, independent of motion or continuity cues that video inputs may introduce. This also avoids overlap with existing video-based spatial benchmarks, and positions our evaluation as a complementary perspective focused purely on spatial reasoning.

To cover different facets of spatial understanding, the benchmark includes tasks such as depth estimation, distance comparison, object relation reasoning, and spatial imagination. Single-view tasks assess inference from a static observation, while multi-view tasks require reasoning across perspectives. A full task list is available in Appx. 7.2, with QA visualizations in Appx. 7.6, and dataset examples shown in Fig. 3.

### 3.4 Data quality check

To ensure the reliability of our benchmark, we conduct a human verification process. Specifically, we sample 400 questions per task in the validation set, forming an 8,000-question benchmark. These questions undergo manual inspection, taking approximately 140 human hours. After filtering out problematic cases, 7,207 questions were retained, resulting in a dataset acceptance rate of 90.1%.

During verification, we focus on identifying vague question stems, incorrect or misleading choices, ambiguous answers, and factual inconsistencies. Errors are marked, discussed, and either corrected or removed. This rigorous process ensures that our benchmark maintains high quality and reliability.

The 90.1% acceptance rate indicates that the vast majority of samples were logically valid, clearly phrased, and visually answerable. In contrast, the 67.27 average human score Tab. 4 reflects the intrinsic difficulty of the proposed spatial reasoning tasks—particularly those involving occlusion, viewpoint ambiguity, or subtle geometric cues. Rather than suggesting flaws in annotation quality, this gap underscores the challenging nature of our benchmark and its ability to expose limitations in both human and model spatial understanding.

### 3.5 3D grounding with VLMs

Expressing an object's spatial position within a scene is a crucial capability. Previous methods primarily relied on point cloud regression or classifying proposal bounding boxes. For video inputs, temproal grounding in VLM [34, 35] is widely explored, [36] use VLM as a frame selector to achieve 3D grounding with a series of tools. In contrast, we propose a novel 3D grounding approach that reformulates this task as a text generation problem, enabling seamless integration with VLMs. Specifically, we transform the 3D grounding task into a frame selection followed by mono-3D grounding. With multi-view input, we first pick a visible frame and infer UV, depth, and size, then recover the 3D box through the camera pose of that frame. While this method already enables the localiza-

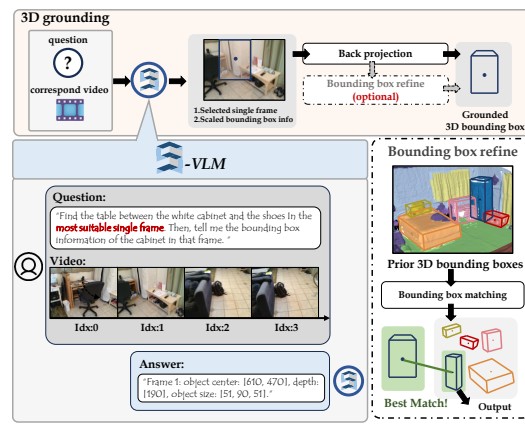

Figure 5: **Our 3D grounding pipeline via VLM.**

tion of a 3D bounding box, we further introduce an optional refinement process. By optimally matching the predicted 3D bounding box with the proposal bounding box, we obtain the final refined result. The overall grounding pipeline is illustrated in Fig. 5.

## 4 Experiments

In this section, we assess the effectiveness of our dataset and investigate the potential upper limits of spatial reasoning in VLMs from the following perspectives: **(i). Performance on existing spatial benchmarks:** we examine how pretraining on *SPAR-7M* improves model performance on image-based spatial benchmarks; **(ii). Evaluation on our in-domain benchmark *SPAR-Bench*:** we examine how pretraining on *SPAR-7M* improves model performance on image-based spatial benchmarks; **(iii). Fine-tuning on 3D-specific tasks:** To investigate the generalization of our approach, we perform supervised fine-tuning (SFT) on 3D-specific benchmarks, including spatial QA and 3D grounding tasks. **(iv) Probing implicit 3D representations:** a BEV coordinate–prediction probe where the model, given multi-view RGB only (no 3D inputs at inference), outputs meter-level object coordinates in an observer-centric frame.

| Methods | VSI-Bench [20] | CV-Bench [37] 2D | CV-Bench [37] 3D | BLINK [18] | 3DSRBench [19] | Seed-Image [21] | MME [39] | MMBench [24] | RealWorldQA [40] | TextVQA [41] |
|---|---|---|---|---|---|---|---|---|---|---|
| GPT-4v [42] | - | 64.3 | 73.8 | 51.14 | - | 71.6 | 1927 | 75.0 | 61.4 | 77.4 |
| GPT-4o [42] | 34.0 | - | - | 60.04 | 45.3 | 77.1 | 2310 | 83.4 | 75.4 | - |
| Cambrain-8B [37] | - | 72.3 | 72.0 | - | - | 74.7 | 1547 | 75.9 | 64.2 | 77.8 |
| LLaVA-OV-7B [4] | 25.3 | - | - | 48.2 | 44.1 | - | 1998 | 80.8 | 66.3 | - |
| InternVL2-8B [43] | 34.6 | - | - | - | - | 75.4 | 2215 | 81.7 | - | 77.4 |
| Base VLM | 32.4 | 74.20 | 78.50 | 46.61 | 58.33 | 76.53 | 2323 | 84.45 | 66.67 | 68.73 |
| +EMOVA-2M | 24.5 | 66.27 | 64.83 | 42.40 | 55.25 | **73.8** | **2186** | **80.24** | 63.14 | **63.78** |
| *+SPAR-mix* | **41.1** | 72.25 | 89.08 | **43.92** | **57.48** | 73.2 | 2163 | 79.90 | **64.71** | 62.91 |

Table 2: **Comparison of model performance on various spatial understanding and 2D general benchmarks.** The base VLM is InternVL2.5-8B, and we use dynamic patch of 3. *Note: Our main comparison is between +EMOVA-2M and +SPAR-mix, as both build upon the same Base VLM and general QA data. For detailed discussion, see section 4.1.*

**Pretraining.** To train a VLM with strong spatial capabilities while preserving generalization, we use a mixed pretraining dataset *SPAR-mix*. 40% of the data is uniformly sampled from our dataset, *SPAR-7M*, while the remaining 60% is uniformly sampled from general 2D data, sourced from the EMOVA configuration. This general data includes tasks such as image-text QA, chart understanding, and OCR. We pretrain the model on a total of 2 million samples, using the base model InternVL2.5-8B. Pretraining was performed on 64 NPUs for 4 days, resulting in a total of about 6,100 NPU-hours.

## 4.1 Evaluation on existing 2D spatial benchmarks

**CV-Bench [37].** The Cambrian Vision-Centric Benchmark (CV-Bench) is a 2,638-sample dataset for evaluating 2D spatial reasoning. It repurposes vision datasets with natural language questions and uses visual prompts (e.g., bounding boxes) to reduce ambiguity and ensure reliable evaluation.

**VSI-Bench [20].** The Visual-Spatial Intelligence Benchmark (VSI-Bench) evaluates spatial reasoning from egocentric video data, containing 5,000+ QA pairs from 288 real-world videos sourced from ScanNet [31], ScanNet++ [32], and ARKitScenes [38]. It covers diverse indoor environments and leverages object-level annotations for question generation. VSI-Bench includes eight tasks across three categories: configurational reasoning (object counting, relative position, route planning), measurement estimation (object and room size, absolute distance), and spatiotemporal reasoning (appearance order in videos). All questions undergo manual review to ensure accuracy.

To evaluate the effectiveness of our dataset, we compare models trained on *SPAR-mix* (ours with EMOVA-2M) against those trained solely on EMOVA-2M. Since our dataset focuses specifically on spatial perception and reasoning, we incorporate a subset of QA data from EMOVA to maintain performance on general-purpose vision-language tasks. Note that the Base VLM has already been pretrained on extensive data with optimized training strategies, making it an unfair baseline for assessing the incremental gain. Therefore, we consider Base VLM + EMOVA-2M as the proper baseline. These results validate the effectiveness of our dataset design and indicate that spatial supervision can be introduced without compromising general QA performance. As shown in Tab. 2, our model achieves state-of-the-art results on CV-Bench 3D and VSI-Bench, demonstrating significant improvements in spatial understanding. While there are minor drops on general benchmarks such as MMBench and TextVQA—mostly within 1%—this is expected, as our data emphasizes spatial reasoning rather than fine-grained vision-language alignment.

**Out-of-domain generalization ability.** We also find that pretraining with our dataset enhances the model's spatial understanding across diverse, non-homogeneous datasets. Although our dataset is primarily derived from indoor scene datasets such as ScanNet [31], ScanNet++ [32], and Structured3D [33], the pretrained model shows consistent improvements across all tasks constructed with datasets different from ours in CV-Bench 3D, as shown in Tab. 3.

| QA source | Hypersim [44] | SUNRGBD [45] | nuScenes [46] |
|---|---|---|---|
| Base VLM | 73 | 87 | 62 |
| *+SPAR-mix* | **93** | **96** | **80** |

Table 3: **Performance improvement on out-of-domain datasets in CV-Bench 3D.**

Notably, our method achieves an 18-point gain on the outdoor nuScenes dataset, highlighting strong generalization beyond indoor scenes. This gain stems from spatial reasoning emphasized in our data—many CV-Bench 3D questions involve relative distance, similar to DistI-OO and ObjRel. Since

these tasks rely on 3D geometric cues rather than appearance or domain-specific semantics, the learned spatial priors transfer effectively across diverse environments.

| Methods | Rank | Avg. | Low | Depth-OC | Depth-OC-MV | Depth-OO | Depth-OO-MV | Dist-OC | Dist-OC-MV | Dist-OO | Dist-OO-MV | Medium | PosMatch | CamMotion | ViewChgI | High | Dist-OO | Dist-OO-MV | ObjRel-OC-MV | ObjRel-OO | ObjRel-OO-MV | SpImg-OC | SpImg-OC-MV | SpImg-OO | SpImg-OO-MV |
|---|---|---|---|---|---|---|---|---|---|---|---|---|---|---|---|---|---|---|---|---|---|---|---|---|---|
| **Baseline** | | | | | | | | | | | | | | | | | | | | | | | | | |
| Chance Level (Random) | - | - | - | | | | | | | | | - | 22.65 | 24.50 | - | 25.09 | 23.82 | 22.02 | 31.25 | 25.27 | 22.16 | 25.81 | 24.42 | 24.17 | 26.89 |
| Chance Level (Frequency) | - | 32.74 | 31.19 | 43.09 | 43.51 | 17.38 | 13.05 | 41.90 | 30.99 | 27.40 | 32.17 | 38.25 | 29.01 | 26.75 | 59.00 | 32.29 | 52.94 | 50.60 | 28.25 | 26.92 | 26.59 | 26.34 | 26.74 | 26.49 | 25.77 |
| ***SPAR-Bench (tiny) API*** | | | | | | | | | | | | | | | | | | | | | | | | | |
| Human Level | 1 | 67.27 | 55.31 | 72.75 | 74.25 | 28.75 | 36.25 | 78.25 | 52.25 | 66.5 | 33.50 | 72.32 | 92 | 64 | 60.97 | 76.22 | 80 | 94 | 70 | 92 | 80 | 78 | 82 | 50 | 60 |
| GPT-4o [42] | 3 | 36.39 | 29.25 | 53.80 | 45.00 | 15.00 | 13.60 | 37.40 | 34.40 | 23.40 | 24.40 | 24.93 | 30 | 16 | 28.80 | 45.11 | 64 | 64 | 58 | 46 | 46 | 32 | 44 | 30 | 22 |
| Claude-3.7-Sonnet [47] | 5 | 21.77 | 25.43 | 41.00 | 45.40 | 11.20 | 12.20 | 42.60 | 19.60 | 26.00 | 5.40 | 7.33 | 16 | 6 | 0.00 | 23.33 | 40 | 48 | 22 | 36 | 14 | 12 | 20 | 6 | 12 |
| Qwen2-VL-72B [48] | 4 | 35.62 | 35.28 | 45.40 | 49.80 | 13.80 | 10.00 | 54.60 | 49.40 | 36.80 | 22.40 | 23.39 | 42 | 18 | 10.16 | 40.00 | 60 | 68 | 50 | 38 | 44 | 18 | 28 | 18 | 36 |
| Qwen2.5-VL-72B [49] | 2 | 39.40 | 35.35 | 53.20 | 46.80 | 17.80 | 29.00 | 49.60 | 57.40 | 14.40 | 14.60 | 23.05 | 40 | 16 | 13.16 | 48.44 | 74 | 74 | 60 | 56 | 50 | 20 | 34 | 24 | 44 |
| ***SPAR-Bench (full)*** | | | | | | | | | | | | | | | | | | | | | | | | | |
| GPT-4o [42] | 3 | 38.11 | 36.88 | 51.22 | 44.69 | 21.21 | 19.33 | 41.40 | 44.90 | 36.34 | 35.96 | 26.49 | 27.74 | 25.25 | 19.99 | 43.80 | 65.00 | 64.88 | 44.75 | 50.82 | 43.21 | 29.84 | 32.56 | 27.81 | 35.29 |
| GPT-4.1 [42] | 2 | 41.60 | 41.95 | 48.25 | 42.66 | 20.86 | 23.58 | 58.29 | 52.33 | 48.65 | 40.44 | 44.02 | 59.29 | 28.75 | 22.02 | 42.93 | 71.76 | 67.26 | 46.25 | 54.95 | 41.00 | 30.38 | 29.65 | 20.20 | 24.93 |
| Doubao-1.5-vision-pro [50] | 1 | 41.74 | 33.24 | 34.89 | 37.22 | 25.24 | 21.40 | 35.04 | 34.99 | 40.58 | 36.57 | 47.36 | 55.73 | 39.00 | 29.47 | 49.49 | 74.71 | 69.64 | 35.75 | 70.33 | 47.65 | 34.95 | 33.14 | 35.76 | 42.86 |
| InternVL2-2B [43] | 19 | 28.06 | 21.74 | 18.06 | 24.81 | 23.20 | 20.97 | 19.47 | 19.95 | 26.83 | 20.61 | 22.83 | 39.69 | 23.00 | 5.81 | 35.42 | 51.18 | 55.95 | 46.00 | 31.59 | 23.82 | 36.02 | 34.30 | 17.55 | 22.41 |
| InternVL2-4B [43] | 12 | 32.01 | 28.94 | 23.94 | 27.22 | 20.00 | 18.12 | 42.57 | 40.16 | 31.29 | 28.18 | 29.16 | 49.87 | 21.00 | 16.62 | 35.70 | 56.76 | 55.36 | 40.25 | 36.81 | 25.21 | 28.76 | 32.27 | 21.19 | 24.65 |
| InternVL2-8B [43] | 11 | 33.02 | 26.83 | 25.75 | 30.88 | 20.67 | 20.78 | 39.03 | 36.19 | 19.15 | 22.19 | 36.49 | 63.36 | 28.00 | 18.11 | 37.37 | 64.71 | 54.46 | 42.75 | 37.36 | 26.32 | 34.14 | 31.10 | 20.86 | 24.65 |
| InternVL2.5-2B [51] | 16 | 30.14 | 25.79 | 39.67 | 39.72 | 12.12 | 15.03 | 30.94 | 29.59 | 20.22 | 19.02 | 22.93 | 37.91 | 24.25 | 6.64 | 36.41 | 51.47 | 56.85 | 50.25 | 33.79 | 24.10 | 27.15 | 35.17 | 26.49 | 22.41 |
| InternVL2.5-4B [51] | 15 | 30.55 | 25.66 | 29.06 | 32.97 | 21.77 | 16.83 | 20.84 | 26.85 | 28.13 | 28.79 | 29.75 | 47.07 | 33.25 | 8.92 | 35.16 | 54.12 | 58.93 | 35.50 | 29.67 | 34.63 | 24.73 | 31.39 | 19.21 | 28.29 |
| InternVL2.5-8B [51] | 6 | 36.28 | 29.46 | 25.78 | 29.31 | 23.79 | 18.76 | 46.82 | 42.68 | 22.62 | 25.89 | 31.88 | 61.32 | 28.00 | 6.32 | 43.80 | 59.71 | 56.85 | 51.75 | 44.23 | 41.55 | 36.56 | 41.57 | 22.52 | 39.50 |
| InternVL2.5-26B [51] | 7 | 34.11 | 24.18 | 41.83 | 36.28 | 18.09 | 17.96 | 22.62 | 17.97 | 22.01 | 16.63 | 50.11 | 69.47 | 30.75 | 7.03 | 42.40 | 62.94 | 58.33 | 45.25 | 54.67 | 50.97 | 24.46 | 25.00 | 27.48 | 32.49 |
| InternVL2.5-38B [51] | 8 | 33.83 | 26.00 | 42.03 | 38.81 | 17.10 | 17.77 | 17.58 | 17.84 | 29.37 | 27.51 | 30.89 | 44.53 | 17.25 | 9.70 | 44.13 | 69.12 | 66.67 | 43.75 | 64.29 | 37.67 | 25.27 | 31.98 | 31.79 | 26.61 |
| LLaVA-OV-0.5B [4] | 17 | 29.48 | 30.14 | 49.22 | 42.72 | 18.04 | 14.92 | 31.48 | 25.67 | 28.98 | 30.10 | 15.89 | 24.43 | 21.75 | 1.50 | 33.42 | 50.88 | 50.00 | 32.00 | 27.75 | 26.04 | 30.91 | 34.01 | 24.50 | 24.65 |
| LLaVA-OV-7B [4] | 13 | 31.20 | 21.79 | 30.33 | 26.94 | 18.58 | 13.87 | 10.43 | 13.64 | 31.24 | 29.29 | 26.13 | 38.68 | 30.25 | 9.47 | 40.14 | 56.47 | 55.06 | 37.25 | 48.63 | 38.23 | 30.38 | 33.72 | 26.49 | 35.01 |
| Qwen2-VL-2B [48] | 20 | 24.60 | 19.43 | 38.03 | 40.63 | 18.84 | 14.09 | 7.81 | 7.07 | 17.82 | 11.14 | 27.55 | 26.21 | 25.25 | 31.20 | 28.22 | 54.12 | 49.11 | 21.75 | 25.27 | 12.47 | 23.92 | 27.62 | 24.83 | 14.85 |
| Qwen2-VL-7B [48] | 14 | 30.74 | 27.52 | 35.97 | 35.22 | 20.83 | 12.88 | 28.68 | 29.95 | 28.21 | 28.45 | 20.44 | 35.37 | 20.25 | 5.69 | 37.03 | 59.71 | 52.38 | 30.25 | 38.46 | 41.00 | 22.04 | 28.49 | 22.52 | 38.38 |
| Qwen2.5-VL-7B [49] | 10 | 33.07 | 28.75 | 31.33 | 33.66 | 21.99 | 14.97 | 42.88 | 37.73 | 23.83 | 23.64 | 22.97 | 33.33 | 28.75 | 6.83 | 40.27 | 58.24 | 51.49 | 44.75 | 50.00 | 32.13 | 33.87 | 32.85 | 27.15 | 31.93 |
| Qwen2.5-VL-32B [49] | 9 | 33.09 | 27.09 | 37.47 | 35.25 | 15.91 | 15.75 | 34.20 | 33.01 | 26.20 | 18.96 | 33.92 | 34.10 | 33.75 | 13.15 | 40.44 | 58.82 | 61.31 | 37.75 | 51.10 | 34.35 | 26.08 | 33.14 | 25.83 | 35.57 |
| Qwen2.5-VL-72B [49] | 5 | 37.01 | 29.94 | 37.47 | 43.00 | 19.52 | 18.36 | 38.72 | 36.44 | 27.80 | 18.25 | 44.61 | 56.49 | 32.75 | 17.27 | 43.80 | 58.82 | 61.90 | 40.75 | 53.57 | 45.98 | 26.88 | 35.17 | 34.11 | 36.97 |
| LLaVA-v1.5-7B [52] | 21 | 23.65 | 10.85 | 5.17 | 12.53 | 17.37 | 11.34 | 7.25 | 5.26 | 18.73 | 9.12 | 26.50 | 24.03 | 28.31 | 34.09 | 34.09 | 51.18 | 52.38 | 34.25 | 24.18 | 26.87 | 34.68 | 29.94 | 22.52 | 30.81 |
| LLaVA-v1.5-13B [52] | 18 | 28.62 | 25.92 | 36.28 | 32.84 | 9.06 | 9.41 | 33.72 | 31.07 | 30.85 | 24.11 | 31.03 | 31.55 | 30.50 | 12.10 | 32.33 | 54.71 | 50.60 | 46.25 | 20.60 | 23.82 | 25.54 | 25.87 | 17.22 | 26.33 |
| LLaVA-v1.6-7B [53] | 22 | 13.21 | 8.53 | 12.14 | 0.00 | 20.35 | 0.27 | 10.76 | 0.41 | 24.27 | 0.00 | 4.79 | 6.62 | 7.75 | 0.00 | 20.18 | 51.76 | 7.74 | 6.25 | 32.14 | 6.37 | 39.52 | 10.47 | 21.52 | 5.88 |
| SpaceR-7B [54] | 4 | 37.55 | 30.90 | 36.14 | 35.16 | 15.40 | 19.25 | 40.13 | 35.07 | 36.97 | 29.16 | 31.93 | 38.68 | 40.00 | 17.23 | 45.61 | 62.35 | 61.61 | 52.25 | 51.92 | 46.81 | 37.90 | 36.05 | 24.83 | 34.17 |
| InternVL2.5-8B+*SPAR-mix* | - | 63.25 | 65.53 | 81.53 | 79.22 | 38.25 | 35.51 | 78.93 | 79.18 | 68.13 | 63.50 | 63.01 | 78.88 | 73.00 | 37.14 | 61.29 | 86.47 | 79.76 | 64.00 | 69.23 | 59.00 | 47.31 | 50.00 | 42.38 | 53.50 |

Table 4: **Performance of different models on *SPAR-Bench*.** Shaded cells indicate the best scores in each category. The highest, second-highest, and third-highest scores in each category are highlighted with light red , light orange , and light yellow , respectively. *SPAR-Bench* (*tiny*) refers to a subset of the full benchmark, where 50 questions are sampled per task.

## 4.2 Evaluation on *SPAR-Bench*

Our model performs well on out-of-domain spatial benchmarks. We further evaluate it on our *SPAR-Bench* and assess the spatial capabilities of other large language models.

**Human vs. model performance.** Tab. 4 shows that human performance consistently exceeds all models across difficulty levels, with a clear margin. While the strongest baseline (Qwen2.5-VL-72B) achieves competitive results, it still falls far short of human-level performance—particularly in tasks requiring spatial reasoning across views or precise geometric estimation.

While performance across levels is not directly comparable due to different metrics (MRA vs. accuracy), comparing to human results reveals a clear trend: models perform moderately on low-level tasks (e.g., 35.4% for Qwen2.5-VL-72B), but drop significantly on middle-level tasks (23.1%), indicating difficulty in structured spatial reasoning. Performance partially rebounds on high-level tasks (48.4%), likely aided by language priors or pattern-based inference. Tasks involving viewpoint changes, spatial imagination, or indirect metric reasoning remain the most challenging, underscoring current limits in spatial generalization and multi-view consistency.

**Impact of data augmentation.** Fine-tuning InternVL2.5-8B on *SPAR-mix* significantly boosts performance across all difficulty levels, reaching 65.53 (low), 63.01 (mid), and 61.29 (high). Gains are largest in depth and multi-view reasoning (e.g., View Change Inference: 6.32→37.14), indicating better cross-view geometric alignment. Notably, the fine-tuned model surpasses human performance in depth prediction, and even the strongest pre-finetuning model exceeds humans on distance estimation. This is understandable, as these metrics are MRA-based and favor numerically optimized models over human intuition.

These results confirm that our dataset introduces valuable inductive bias for spatial reasoning. However, gains remain uneven, and a clear gap persists compared to human-level performance—highlighting the need for advances in multi-view fusion, spatial representation learning, and compositional reasoning.

| | | SQA3D [14] | ScanQA [13] | | |
|---|---|---|---|---|---|
| Methods | 3D | EM | BLEU-4 | CiDEr | EM |
| 3D-LLM [8] | ✓ | - | 12.0 | 69.4 | 20.4 |
| Chat-3D v2 [55] | ✓ | 54.7 | 14.0 | 87.6 | - |
| LEO [15] | ✓ | 50.0 | 13.2 | 101.4 | 21.5 |
| LL3DA [9] | ✓ | - | 13.5 | 76.8 | - |
| Scene-LLM [12] | ✓ | 54.2 | 12.0 | 80.0 | 27.2 |
| LLaVA-3D [10] | ✓ | 55.6 | 14.5 | 91.7 | 27.0 |
| Video-3D LLM [56] | ✓ | **58.6** | **16.2** | **102.1** | **30.1** |
| *SPAR-mix* | ✗ | 58.1 | 15.3 | 90.7 | 27.7 |

| Methods | Acc@0.25 | Acc@0.5 |
|---|---|---|
| ScanRefer [57] | 37.3 | 24.3 |
| MVT [58] | 40.8 | 33.3 |
| ViL3DRel [59] | 47.9 | 37.7 |
| 3D-LLM [8] | 30.3 | - |
| Chat-3D v2 [55] | 35.9 | 30.4 |
| Grounded 3D-LLM [60] | 47.9 | 44.1 |
| LLaVA-3D [10] | 54.1 | 42.4 |
| Video-3D LLM [56] | **58.1** | **51.7** |
| *SPAR-mix* | 48.8 (31.9) | 43.1 (12.4) |

Table 5: **Comparison of SQA3D and ScanQA performance.** "3D" indicates whether the model is infused with 3D information.

Table 6: **Comparison of ScanRefer performance across different models.** The content in "()" indicates results without refine.

## 4.3 Evaluation on 3D-specific tasks

In addition to the previously discussed spatial benchmarks, there are also tasks specifically defined in 3D scenes, which typically involve questions related to the entire scene. After pretraining the VLM using our dataset, we perform supervised fine-tuning (SFT) on 3D tasks to further investigate the spatial capabilities of the VLM.

**Spatial QA.** We use two benchmarks: ScanQA, which focuses on object attributes and spatial relationships within a scene, and SQA3D, a spatial benchmark where the task is to answer specific questions based on a given viewpoint description within the scene. This requires the model to demonstrate spatial imagination. We treat this as a multi-view task during training. As shown in Tab. 5, despite not incorporating 3D information into the model, it still achieves the competitive scores when compared to the 3D-enhanced LLMs.

**3D grouding.** We evaluate the 3D grounding capability of our model using the ScanRefer dataset. ScanRefer requires the model to locate objects in a scene based on textual descriptions. As shown in the Tab. 6, the values in parentheses represent the performance without proposal refinement. We observed that the VLM's depth predictions may sometimes be inaccurate when the distance is large due to the absence of 3D information. However, after refining, the results perform well. These results underline the potential of leveraging a VLM for grounding, showcasing its ability to perform well without the need for 3D information, while also tackling a more complex generative task.

## 4.4 Probing Implicit 3D Representations

We probe whether the model learns implicit 3D structure beyond QA metrics. Following language-guided cognitive maps [20], the model predicts bird's-eye-view (BEV) 2D coordinates for multiple objects from multi-view RGB only, with no 3D supervision at inference. Each instance uses an observer-centric frame anchored at the first view: origin at the camera center pro-

| Model | APE Mean ↓ | APE P50 ↓ | APE P90 ↓ |
|---|---|---|---|
| Base VLM | 2.22 | 1.86 | 4.12 |
| *SPAR-mix* | **1.05** | **0.64** | **2.45** |

Table 7: **BEV coordinate prediction on Scan-Net++ val.** *SPAR-mix* reduces spatial error.

jected to the ground plane, the y-axis along the viewing direction, and the x-axis to the observer's right. The prompt states these rules and asks for meter-level coordinates in a fixed format.

We build a validation set of 1,000 ScanNet++ samples, each with three images and 3–7 objects with known 3D positions. We evaluate object-level Average Position Error (APE) and compare our *SPAR-mix* model with a 2D-only base VLM (InternVL2.5-8B).

*SPAR-mix* shows lower localization error than the base model. By binning objects by distance from the main view, *SPAR-mix* consistently outperforms the base model, with larger gains at medium–far ranges where depth and occlusion are harder (e.g., mean APE: 1.46 vs. 3.66 at 3–5 m; 2.41 vs. 5.62 at 5–7 m; 4.43 vs. 7.62 at 7–10 m).

These results suggest that training with *SPAR-mix* yields a scene-consistent internal layout that aggregates cues across views to infer geometry, even without explicit 3D input at inference.

| Distance (m) | #Objs | APE Mean ↓ | | APE P90 ↓ | |
| --- | --- | --- | --- | --- | --- |
| | | Base VLM | *SPAR-mix* | Base VLM | *SPAR-mix* |
| [0, 1) | 990 | 0.74 | **0.70** | **1.03** | 1.61 |
| [1, 2) | 2635 | 1.40 | **0.73** | 1.88 | **1.77** |
| [2, 3) | 1619 | 2.39 | **1.04** | 2.86 | **2.25** |
| [3, 5) | 1225 | 3.66 | **1.46** | 4.50 | **3.26** |
| [5, 7) | 346 | 5.62 | **2.41** | 6.55 | **4.64** |
| [7, 10) | 62 | 7.62 | **4.43** | 8.42 | **7.38** |
| [10, ∞) | 5 | 12.71 | **11.57** | 16.46 | **15.26** |

Table 8: **BEV APE across distance bins.** SPAR-mix shows larger gains in medium-to-far ranges.

## 4.5 Ablation study

We evaluate the impact of SPAR data proportions on InternVL2.5-4B, trained with 2M samples from SPAR (3D) and EMOVA (2D), using an unfrozen ViT. We adopt ViT fine-tuning as it improves overall performance and better adapts to the diverse spatial structures in *SPAR-7M*. Tab.9 presents results on CV-Bench (2D, 3D) and SPAR-Bench (Low, Middle, High levels).

| SPAR Ratio | CV-Bench [37] | | SPAR-Bench | | |
| --- | --- | --- | --- | --- | --- |
| | 2D | 3D | Low | Medium | High |
| 40% | **74.13** | 89.09 | 66.97 | 42.21 | 48.19 |
| 60% | 73.00 | **91.10** | 67.01 | **44.04** | **51.86** |

Table 9: **Performance comparison of different SPAR ratios.**

**Effect of SPAR ratio.** With an unfrozen ViT, reducing the *SPAR-7M* ratio from 60% to 40% leads to a slight improvement on CV-Bench 2D (73.00 → 74.13), but results in performance drops on CV-Bench 3D (91.10 → 89.09) and across all levels of *SPAR-Bench*—particularly on the middle (44.04 → 42.21) and high (51.86 → 48.19) tasks. This indicates while 2D benchmarks may benefit from increased exposure to EMOVA-style data, an excessive reduction in the *SPAR-7M* proportion can cause the model to forget generalizable spatial priors. In contrast, complex spatial reasoning—especially under multi-view or relational conditions—relies more heavily on training signals tailored for spatial understanding. Maintaining or increasing the proportion of *SPAR-7M* is thus critical for reinforcing geometric alignment and ensuring spatial consistency across diverse viewpoints.

## 5 Conclusion

We introduce *SPAR-7M*, a large-scale dataset constructed from thousands of scenes, specifically designed to enhance spatial reasoning. To provide a rigorous evaluation framework, we propose *SPAR-Bench*, a benchmark that offers a more comprehensive assessment of spatial understandi ng, supporting both single-view and multi-view inputs. Experimental results show that pretraining on *SPAR-7M* alongside general datasets enables models to achieve state-of-the-art performance on VSI-Bench and CV-Bench 3D. Furthermore, fine-tuning on 3D task-specific datasets yields competitive results, highlighting the effectiveness of our dataset in bridging the gap between 2D and 3D spatial reasoning. While our tasks span diverse spatial challenges, they are primarily static QA-based and do not yet capture continuous spatial reasoning required in embodied or interactive settings.

## 6 Limitation

Our study is largely static and image-centric: QA benchmarks with single/multi-view RGB do not capture continuous, interactive reasoning for embodied or long-horizon video. Without metric 3D, monocular reconstruction suffers scale ambiguity, so metric tasks need extra cues (e.g., known sizes, stereo) or ordinal/normalized targets. Our data pipeline also relies on 3D ground truth—depth, intrinsics, and poses—which may be unavailable in some domains.

## Acknowledgments

This work was supported in part by National Natural Science Foundation of China (Grant No. 62376060).

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

# 7 Appendix section

## 7.1 Inference Efficiency Analysis

| Model | 3D | Feature Extr. (s) | Inference (ms) | Peak Mem. (MB) | CiDEr (ScanQA) |
|---|---|---|---|---|---|
| 3D-LLM [60] | ✓ | 900.0 | **197.4** | 16076.4 | 69.4 |
| LL3DA [9] | ✓ | 0.3 | 382.0 | **3306.4** | 76.8 |
| LEO [15] | ✓ | 4.8 | 505.6 | 15320.0 | **101.4** |
| LLaVA-3D [10] | ✓ | 0.2 | 494.1 | 15313.5 | 91.7 |
| *SPAR-mix* ($13\times448\times448$) | ✗ | 0.14 | 771.1 | 19473.7 | 90.7 |
| *SPAR-mix* ($11\times448\times448$) | ✗ | 0.12 | 662.1 | 18942.8 | 89.6 |
| *SPAR-mix* ($13\times224\times224$) | ✗ | **0.04** | 245.6 | 16982.1 | 87.6 |

Table 10: **Inference efficiency on A6000 (bf16 + FlashAttention).** We report feature extraction latency, end-to-end inference latency, and peak GPU memory [10]. Accuracy is summarized by CiDEr on ScanQA. Our 2D pipeline avoids 3D preprocessing overhead while allowing configurable frame count and resolution, trading sequence length for accuracy and latency.

We evaluate inference efficiency on a single NVIDIA A6000 with bf16 and FlashAttention. We report feature-extraction latency (raw inputs to encoder features), end-to-end inference latency (encoder + LLM), peak GPU memory [10], and CIDEr on ScanQA. Results are in Table 10.

Unlike 3D VLMs, our 2D pipeline needs no point-cloud, voxel, or mesh preprocessing. Feature extraction is mainly image processing (0.14 s for 13×448×448), whereas 3D pipelines range from sub-second to minutes.

Latency and memory scale with the number of vision tokens. Reducing frames or resolution lowers cost with modest accuracy loss; for example, 13×448×448 → 13×224×224 cuts inference from 771.1 ms to 245.6 ms, with CIDEr decreasing from 90.7 to 87.6.

Peak memory can be higher than some 3D models due to many 2D tokens. Overall, our approach provides 3D-free inference with competitive accuracy and simple knobs—frame count and resolution—to meet latency and memory budgets.

## 7.2 Benchmark task descriptions

The tasks are categorized into single-view and multi-view settings, covering depth estimation, distance prediction, spatial relations, and spatial imagination. The descriptions of each task are as follows:

**Single-view tasks** Single-view tasks test a model's ability to infer spatial properties from a single image. These tasks include:

- **Depth estimation (OC, OO, NA)**: Predicting absolute or relative depth values for objects
- **Distance prediction (OC, OO, NA)**: Estimating the Euclidean distance between objects or from an object to the camera.
- **Object center distance inference (OO, MCA)**: Given objects A, B and C, determine which of B and C is farther or closer to A.
- **Object spatial relation (OO, MCA)**: Determining relative positioning (e.g., left, right, in front of).
- **Spatial imagination (OC, OO, MCA)**: Predicting unseen spatial relationships based on limited visual information.

**Multi-view tasks** Multi-view tasks require reasoning across multiple images to infer spatial relationships. These tasks include:

- **Viewpoint change inference (NA)**: Given two perspectives, output how the camera should be moved to see the second perspective.

- **Multi-view depth estimation (OC, OO, NA)**: Predicting depth across multiple perspectives.
- **Multi-view distance prediction (OC, OO, NA)**: Estimating object distances across different views.
- **Multi-view object matching (MCA)**: Identifying the same object across multiple views.
- **Camera pose inference (MCA)**: Predict the position of the camera corresponding to the second perspective in the first image.
- **Multi-view object spatial relation (OC, OO, MCA)**: Determining object relationships across multiple images.
- **Spatial imagination (OC, OO, MCA)**: Reasoning about spatial structure beyond visible views.

A tiny version of our *SPAR-Bench* evaluation results are shown in Tab. 11.

## 7.3 Image subsampling

We propose an efficient image filtering method based on camera poses to reduce redundant images with high similarity, so that can improve data processing efficiency. Given a scene $S$ with a set of image sequence $\mathcal{I}$, our goal is to filter out similar images based on a translation threshold $d_{\text{trans}}$ and a rotation angle threshold $d_{\text{rot}}$, obtaining a compact image sequence $\mathcal{I}' \subseteq \mathcal{I}$.

Specifically, for a given image sequence $\mathcal{I}$, we first load the corresponding camera intrinsic and extrinsic parameters. Each camera pose is represented by a $4 \times 4$ transformation matrix $\mathbf{T}_i$, consisting of a rotation matrix $\mathbf{R}_i$ and a translation vector $\mathbf{t}_i$:

$$\mathbf{T}_i = \begin{bmatrix} \mathbf{R}_i & \mathbf{t}_i \\ \mathbf{0}^T & 1 \end{bmatrix} \tag{1}$$

where $\mathbf{t}_i \in \mathbb{R}^3$ and $\mathbf{R}_i \in \text{SO}(3)$. The world-to-camera transformation by inverting the given pose.

**Translation filtering** For each image $i(i = 1, .., n)$, we compute the Euclidean distance between its translation vector $\mathbf{t}_i$ and the translation vector $\mathbf{t}_j$ of a candidate image $j(j = i + 1, .., n)$:

$$d_{ij}^{trans} = \|\mathbf{t}_i - \mathbf{t}_j\| \tag{2}$$

If $d_{ij}^{trans} > d_{th}$, we believe that the difference between these two frames is significant enough and we will preserve the current frame $j$. If $d_{ij}^{trans} < d_{th}$, we will further perform rotation filtering.

**Rotation filtering** For images with smaller $d_{ij}^{trans}$, we compute the relative rotation matrix: $\mathbf{R}_{ij} = \mathbf{R}_i^{-1}\mathbf{R}_j$ The rotational difference is determined by the angle $\theta_{ij}$, computed as:

$$\theta_{ij} = \cos^{-1}\left(\frac{\text{Trace}(\mathbf{R}_{ij}) - 1}{2}\right) * \frac{180}{\pi} \tag{3}$$

If $\theta_{ij} < \theta_{th}$, image $j$ is considered redundant and removed. After iterating through all images, the final filtered image set is as follows:

$$\mathcal{I}' = \{i \in \mathcal{I} \mid \text{satisfies filtering criteria}\} \tag{4}$$

This method can filter out approximately 90% of redundant images, which ensures that only images with sufficiently distinct poses are retained, reducing redundancy while preserving viewpoint diversity.

In the experimental setup, we set the threshold parameter of the ScanNetPP dataset [32] with $d_{th} = 0.5$ and $\theta_{th} = 45$ and ScanNet [31] dataset with $d_{th} = 0.5$ and $\theta_{th} = 15$. For the Structured3D Dataset, we did not perform filtering and subsampling operations since the images in the dataset were sparse enough.

## 7.4 Image item construction

Given a scene $S$, we construct image items by extracting 3D object data and projecting it onto 2D images.

| Methods | Avg. | Low | Depth-OC | Depth-OC-MV | Depth-OO | Depth-OO-MV | Dist-OC | Dist-OC-MV | Dist-OO | Dist-OO-MV | Medium | PosMatch | CamMotion | ViewChgI | High | Dist-OO | Dist-OO-MV | ObjRel-OC-MV | ObjRel-OO | ObjRel-OO-MV | Spimg-OC | Spimg-OC-MV | Spimg-OO | Spimg-OO-MV |
|---|---|---|---|---|---|---|---|---|---|---|---|---|---|---|---|---|---|---|---|---|---|---|---|---|
| **Baseline** | | | | | | | | | | | | | | | | | | | | | | | | |
| Chance Level (Random) | - | - | - | - | - | - | - | - | - | - | | 22 | 18 | - | | 80 | 32 | 26 | 28 | 22 | 32 | 12 | 30 | 28 |
| Chance Level (Frequency) | 37.80 | 36.33 | 42.89 | 51.78 | 25.78 | 27.11 | 35.33 | 46.89 | 35.33 | 25.56 | 41.14 | 30 | 40 | 53.42 | 38.00 | 60 | 58 | 32 | 30 | 30 | 32 | 34 | 28 | 38 |
| Human Level | 67.27 | 55.31 | 72.75 | 74.25 | 28.75 | 36.25 | 78.25 | 52.25 | 66.5 | 33.50 | 72.32 | 92 | 64 | 60.97 | 76.22 | 80 | 94 | 70 | 92 | 80 | 78 | 82 | 50 | 60 |
| GPT-4o | 36.39 | 29.25 | 53.80 | 45.00 | 15.00 | 13.60 | 37.40 | 34.40 | 23.40 | 24.40 | 24.93 | 30 | 16 | 28.80 | 45.11 | 64 | 64 | 58 | 46 | 46 | 32 | 44 | 30 | 22 |
| Claude-3.7-Sonnet | 21.77 | 25.43 | 41.00 | 45.40 | 11.20 | 12.20 | 42.60 | 19.60 | 26.00 | 5.40 | 7.33 | 16 | 6 | 0.00 | 23.33 | 40 | 48 | 22 | 36 | 14 | 12 | 20 | 6 | 12 |
| Qwen2-VL-72B | 35.62 | 35.28 | 45.40 | 49.80 | 13.80 | 10.00 | 54.60 | 49.40 | 36.80 | 22.40 | 23.39 | 42 | 18 | 10.16 | 40.00 | 60 | 68 | 50 | 38 | 44 | 18 | 28 | 18 | 36 |
| Qwen2.5-VL-72B | 39.40 | 35.35 | 53.20 | 46.80 | 17.80 | 29.00 | 49.60 | 57.40 | 14.40 | 14.60 | 23.05 | 40 | 16 | 13.16 | 48.44 | 74 | 74 | 60 | 56 | 50 | 20 | 34 | 24 | 44 |
| InternVL2-2B | 29.51 | 21.85 | 15.00 | 31.40 | 17.80 | 18.80 | 13.40 | 27.40 | 26.40 | 24.60 | 25.81 | 44 | 26 | 7.44 | 37.56 | 46 | 56 | 54 | 42 | 18 | 50 | 42 | 14 | 16 |
| InternVL2-4B | 32.10 | 29.55 | 22.02 | 28.40 | 18.80 | 14.20 | 47.60 | 52.60 | 26.00 | 26.60 | 33.88 | 52 | 30 | 19.64 | 33.78 | 46 | 54 | 44 | 30 | 30 | 26 | 26 | 26 | 22 |
| InternVL2-8B | 32.95 | 24.10 | 24.60 | 39.00 | 16.00 | 16.80 | 35.40 | 33.40 | 13.40 | 14.20 | 35.43 | 58 | 28 | 20.28 | 40.00 | 68 | 42 | 40 | 46 | 34 | 34 | 46 | 16 | 34 |
| InternVL2.5-2B | 31.81 | 27.85 | 44.80 | 42.20 | 11.20 | 7.00 | 40.20 | 35.40 | 24.20 | 17.80 | 22.48 | 40 | 22 | 5.44 | 38.44 | 68 | 48 | 50 | 48 | 26 | 18 | 38 | 20 | 30 |
| InternVL2.5-4B | 33.99 | 30.38 | 31.20 | 36.20 | 26.20 | 30.00 | 24.20 | 36.40 | 31.40 | 27.40 | 34.27 | 58 | 38 | 6.80 | 37.11 | 48 | 58 | 54 | 40 | 30 | 24 | 42 | 18 | 20 |
| InternVL2.5-8B | 37.27 | 28.38 | 27.40 | 31.80 | 19.60 | 19.00 | 40.40 | 48.80 | 15.00 | 25.00 | 31.47 | 66 | 22 | 6.40 | 47.11 | 58 | 54 | 50 | 52 | 52 | 44 | 58 | 22 | 34 |
| Qwen2-VL-2b | 26.88 | 23.45 | 44.20 | 50.00 | 25.20 | 17.40 | 7.40 | 12.60 | 20.60 | 10.20 | 28.01 | 22 | 24 | 38.04 | 29.56 | 52 | 50 | 20 | 24 | 10 | 40 | 30 | 24 | 16 |
| Qwen2-VL-7b | 32.84 | 27.98 | 37.80 | 36.20 | 23.60 | 7.00 | 28.00 | 31.80 | 31.60 | 27.80 | 16.36 | 26 | 18 | 5.08 | 42.67 | 58 | 54 | 26 | 40 | 54 | 34 | 36 | 40 | 42 |
| Qwen2.5-VL-7b | 33.48 | 31.25 | 27.80 | 37.20 | 27.40 | 19.80 | 50.00 | 47.60 | 17.60 | 22.60 | 19.84 | 26 | 24 | 9.52 | 40.00 | 52 | 50 | 44 | 56 | 28 | 28 | 36 | 32 | 34 |
| LLaVA-OV-0.5B | 30.84 | 33.20 | 55.40 | 51.60 | 22.80 | 10.00 | 35.20 | 28.20 | 36.60 | 25.80 | 15.08 | 24 | 22 | 1.24 | 34.00 | 52 | 56 | 40 | 36 | 16 | 30 | 40 | 22 | 14 |
| LLaVA-OV-7B | 34.73 | 27.95 | 42.80 | 44.60 | 25.20 | 24.00 | 12.80 | 12.60 | 38.40 | 23.20 | 27.69 | 48 | 22 | 13.08 | 43.11 | 64 | 62 | 26 | 58 | 42 | 24 | 40 | 32 | 40 |
| llava-v1.5-7b | 25.76 | 13.02 | 4.80 | 15.40 | 17.60 | 17.60 | 8.80 | 7.80 | 17.60 | 14.60 | 33.69 | 28 | 40 | 33.08 | 34.44 | 52 | 54 | 18 | 22 | 26 | 42 | 38 | 18 | 40 |
| llava-v1.6-7b | 13.50 | 9.00 | 10.60 | 0.00 | 20.40 | 0.00 | 16.20 | 0.00 | 24.80 | 0.00 | 6.00 | 8 | 10 | 0.00 | 20.00 | 46 | 14 | 12 | 30 | 6 | 42 | 6 | 20 | 4 |
| ours | 66.65 | 70.33 | 87.00 | 83.20 | 45.80 | 43.20 | 81.00 | 84.00 | 78.80 | 59.60 | 60.13 | 78 | 66 | 36.40 | 65.56 | 86 | 90 | 72 | 78 | 58 | 48 | 48 | 42 | 68 |

Table 11: **Performance of different models on *SPAR-Bench*.** All results are obtained on tiny *SPAR-Bench*. Shaded cells indicate best scores in each category.

**Data loading and initialization** For each scene, we load the corresponding 3D mesh, camera intrinsic and extrinsic parameters, and instance annotations. The scene mesh is represented as: $M = (\mathcal{V}, \mathcal{F})$ where $\mathcal{V}$ is the set of vertices and $\mathcal{F}$ is the set of triangular faces.

To determine the visibility of 3D faces in the image, we perform rasterization to obtain a mapping from image pixels to face indices: $pix\_to\_face_{(x,y)} = f_k$. where $f_k \in \mathcal{F}$ and $pix\_to\_face$ stores the corresponding face index for each pixel $(x, y)$. If a pixel does not correspond to any face, it will be marked as -1.

**Object projection and bounding box computation** For each 3D object, we compute the set of visible vertices and project them into the 2D image plane using:

$$p_{2D} = K(R \cdot p_{3D} + t) \tag{5}$$

where $p_{3D}$ is a vertex in the 3D space, and $p_{2D}$ is its projected 2D coordinate. The bounding box of the projected object is computed as: $B_{obj} = [x_{min}, y_{min}, x_{max}, y_{max}]$. To ensure a valid projection, we also enforce some constraints as follows:

- The fraction of visible object vertices $f_v$ in the image must exceed a threshold $\tau_v$, where $f_v = \frac{|V_{visible}|}{|V_{total}|}$.
- The projected object area must be above a minimum threshold $A_{min}$.
- The depth values in the z-buffer must be within a reasonable range. where $z_{min} = \min(Z_{obj})$, $z_{max} = \max(Z_{obj})$

Each 3D object is associated with an oriented bounding box, defined by its centroid $c$, axis-aligned lengths $l_x, l_y, l_z$. Finally, the extracted image item dictionary, including object data, is used for downstream task generation.

### 7.5 Task data generation

In this section, we describe the detailed information on multi-task generation. We generate questions based on the template. These questions can be of three types: select, fill, or sentence. In each case, the goal is to generate a question that involves the spatial relationship between two objects. We will provide a Q&A format in the form of a template and fill in key information and answers in it.

**Obj spatial relation** This task is to describe the spatial relationships between objects in the 3D scene based on their spatial positions. The process involves several key steps: (1) Transforming 3D

object coordinates from the original camera view into a common view. This transformation ensures that all spatial calculations are relative to the main camera view. Let $c$ denotes the 3D center of an object in the world coordinate system, and $\mathbf{T}$ denotes the camera pose. The transformation is carried out as: $c'_{homo} = \mathbf{T}^{-1} \cdot c_{homo}$. where $c_{homo}$ means homogeneous coordinate of object 3D center point. (2) Spatial Relationship Description. We describe their spatial relationships in terms of several key factors: above-below, left-right, near-far, and front-behind (relative to two objects). These relationships are determined based on their spatial coordinates and distance from the camera center. The distance is calculated by $d = \|c' - T_{trans}\|_2$. We set the relationship threshold at 0.1m. If the difference in coordinates or distances is less than 0.1m, we consider the corresponding spatial relationship to be indistinguishable (empty).

**Depth prediction** Given an image $I$ containing a set of detected objects $\mathcal{O} = \{o_1, o_2, ..., o_n\}$, we transform each object's 3D center point $c$ into the camera coordinate system as $c' \in \mathbb{R}^3$. Then the transformed depth values $d$ are extracted from the $z$-component of their transformed coordinates $c'$, which means $d = c'(z)$.

For the absolute depth prediction task, we use this value as the standard answer. For the relative depth estimation task, we calculate the depth difference between objects by: $\Delta d = |d_i - d_j|$. We will skip that case if two objects have overlapping bounding boxes or similar values.

**Distance infer** Given an image $I$ containing a set of objects $\mathcal{O} = \{o_1, o_2, ..., o_n\}$, we define the 3D center of each object $o_i$ in the world coordinate system as $\mathbf{c}_i \in \mathbb{R}^3$ and transformed them into the camera coordinate system as $\mathbf{c}'_i$. For the object-object type task, we random sample two objects $o_A$ with $\mathbf{c}'_A$ and $o_B$ with $\mathbf{c}'_B$ in the same scene. The Euclidean distance between them is given by: $d_{AB} = \|\mathbf{c}'_A - \mathbf{c}'_B\|_2$, where $\|\cdot\|_2$ represents the $L_2$-norm. For the object-camera type task, we calculate the distance with $|\mathbf{c}'_i|_2$. To ensure numerical stability and consistency in question-answer generation, the computed distance is rounded to the nearest 0.1 meter. If the two objects have overlapping 3D bounding boxes or the distance is smaller than the threshold, we will skip that case.

**Spatial volume infer** For an object $o_i$ in the image, we first obtain its 3D bounding box in the world coordinate system and then transform it into the camera coordinate system by the extrinsic transformation matrix. The center coordinate is denoted as $\mathbf{c}'_i$ and each corner point of 3D bounding box is denoted as $\mathbf{b}'_{ij}$, $j = (1, 2, ..., 8)$. The object's dimensions(length, width, and height) are derived as follows:

$$h = \max_j b^{(z)}_{i,j} - \min_j b^{(z)}_{i,j}$$
$$l = \max_{j,k} \|b^{(xy)}_{i,j} - b^{(xy)}_{i,k}\| \tag{6}$$
$$w = \min_{j,k} \|b^{(xy)}_{i,j} - b^{(xy)}_{i,k}\|$$

where $b^{(xy)}_{i,j} = (b^{(x)}_{i,j}, b^{(y)}_{i,j})$ represents the 2D projection of the bounding box in the XY plane. To ensure consistency, all dimensions are converted to centimeters. The final estimated volume is given by: $V = h \cdot l \cdot w$.

**Spatial imagination** Our spatial imagination task aims to evaluate the spatial reasoning capabilities of LLM models by analyzing object relationships before and after camera transformations in a 3D environment. Given an image and corresponding scene metadata, we randomly sample objects and generate structured question-answer (QA) pairs that describe spatial relationships.

For each image $I$, it is associated with a set of objects $\mathcal{O} = \{o_1, o_2, ..., o_N\}$. We randomly sample objects $o_A, o_B, o_C, o_D \subset \mathcal{O}$ for relational comparisons. To analyze object relationships from different viewpoints, we transform the camera pose $\mathbf{P} \in SE(3)$ so that it moves towards object $A$ and faces object $B$. The new camera pose is constructed as follows:

$$\mathbf{t}_{A \to B} = \mathbf{c}_A \quad \mathbf{f} = \frac{\mathbf{c}_B - \mathbf{c}_A}{\|\mathbf{c}_B - \mathbf{c}_A\|}$$
$$\mathbf{v} = \frac{\mathbf{u}_0 \times \mathbf{f}}{\|\mathbf{u}_0 \times \mathbf{f}\|} \quad \mathbf{u} = \mathbf{f} \times \mathbf{v} \quad \mathbf{r} = -\mathbf{v} \tag{7}$$

where $\mathbf{c}_A, \mathbf{c}_B$ are 3D center coordinates of object $o_A$ and $o_B$ respectively. $\mathbf{u}_0 = [0, 0, 1]^T$. $\mathbf{f}$ is forward direction vector. $\mathbf{v}, \mathbf{u}, \mathbf{r}$ are left-vector, up-vector and right-vector respectively. So the new

camera pose is computed as:

$$\mathbf{P}_{A \to B} = \begin{bmatrix} \mathbf{r} & \mathbf{u} & \mathbf{f} & \mathbf{t}_{A \to B} \\ 0 & 0 & 0 & 1 \end{bmatrix} \tag{8}$$

We extract the up-vector $\mathbf{u}$ from $\mathbf{P}_{A \to B}$ to compute the vertical rotation angle: $\theta = \cos^{-1}(\mathbf{u}_z)$. If $\theta > 60°$, we will discard this viewpoint to maintain a reasonable observation angle.

After that, we compute the spatial relationship between object $C$ and object $D$ before and after camera transformation. The relationships are determined based on their relative positions in the original and transformed coordinate system. We describe their spatial relationships in terms of several key factors: above-below, left-right, near-far, and front-behind (relative to two objects). Please refer to the paragraph **Obj spatial relation** for details. The final step is to generate structured question-answer pairs. The same procedure is also applied after the camera-object type task.

**Position matching** Position matching aims to identify and compare the positions of the same object across different views. Given an object detected in multiple images, the task is to find its corresponding 2D bounding box in another view based on its known location in one reference image.

We define the set of detected objects as: $\mathcal{O} = \{o_1, o_2, \ldots, o_N\}$. Each object $o_i$ appears in a set of images: $\mathcal{I}_i = \{I_{i_1}, I_{i_2}, \ldots, I_{i_m}\}$. If $m < 2$, the object is discarded. For each valid object, we randomly select two distinct images $I_{i_A}, I_{i_B} \in \mathcal{I}_i$ as a reference frame and target frame. The 3D bounding box of object $o_i$ in world coordinate is denoted as $b_i^{3D}$. We project the 3D bounding box into the image plane by Eq. 5. The 2D bounding boxes in different images are denoted as $b_{i_A}^{2D}, b_{i_B}^{2D}$.

The system formulates the position-matching question as: Qustion = Given object $o_i$ and 2D bounding box $b_{i_A}^{2D}$ in image $I_A$, find its location in image $I_B$; Answer = $b_{i_B}^{2D}$.

**View chang infer** The view change inference task aims to determine the spatial displacement between different images. To ensure images exhibit a co-visibility relationship, we select different perspectives images containing the same object instance.

For two distinct images $I_A, I_B$ containing the same object $o$, we first compute the center of the 2D bounding box to quantify the object's displacement in image space. If 2D bounding box of object $o$ in image $I_A$ is denoted as $b_A^{2D} = [x_A^{\min}, y_A^{\min}, x_A^{\max}, y_A^{\max}]$, the 2D center coordinate can be calculated as $\mathbf{c}_A^{2D} = ((x_A^{\min} + x_A^{\max})/2, (y_A^{\min} + y_A^{\max})/2)$. We determined the object's location in image $I_A, I_B$ according to $\mathbf{c}_A^{2D}$ and $\mathbf{c}_B^{2D}$.

To further analyze the view movement in the world coordinate system, We compute the relative pose transformation: $\mathbf{T}_{AB} = \mathbf{T}_A^{-1}\mathbf{T}_B$. Finally, we determine the translation distance and rotation angle of the viewpoint based on the relative pose transformation matrix and designed rules.

**Camera pose** The camera pose task aims to estimate the relative camera motion and generate questions about the 2D coordinate and depth information.

We select distinct frames $I_A, I_B$ containing the same object $o$ to ensure there is an overlap between them. Firstly, we project the camera position of image $I_B$ in the world coordinate into the coordinate system of image $I_A$. According to Eq. 5, the projected point coordinate is given by $p_{B|A} = K(R_A \cdot p_B + t_A)$, where $R_A$ and $t_A$ are the rotation matrix and the translation vector of $I_A$, K is camera intrinsic parameters. Then we calculate 2D image coordinates $(u, v)$ and normalize to 0-1000:

$$u = \frac{p_{B|A}[0]}{p_{B|A}[2]} \cdot \frac{1000}{\text{width}} \qquad v = \frac{p_{B|A}[1]}{p_{B|A}[2]} \cdot \frac{1000}{\text{height}} \tag{9}$$

If the projected point lies within the bounds of the image $I_A$, we compute the depth as: $d_{B|A} = (R_A \cdot p_B + t_A)[2]$

**Obj frame location** We introduce the object frame location task to identify the frames in which a given object appears. We select a reference frame and determine in which other frames the object is present. This process enables the automatic generation of question-answer pairs related to object appearance across frames.

For an object $o$, we extract the set of frames: $I_o = \{I_1, I_2, ..., I_n\}$ in which it appears. To generate questions and answers for the object frame localization task, we randomly select one frame $I_s$ as the reference frame. We also add some irrelevant frames as wrong options, and the answer consists of the list of other frame indices $I_o \setminus \{I_s\}$ where the object appears.

**Obj frame location** This task infers the chronological order in which multiple objects appear within a sequence of frames. By selecting a subset of objects and analyzing their first occurrence across frames, we generate structured question-answer pairs that facilitate the temporal reasoning ability of LLMs.

Given object set $\mathcal{O} = \{o_1, o_2, ..., o_N\}$ and associated image set $\mathcal{L} = \{\mathcal{I}_1, \mathcal{I}_2, ..., \mathcal{I}_N\}$, where $\mathcal{I}_i$ means the set of frames that object $o_i$ appears, we extract the first appearance frame of each object as $F_i = \min(f \mid f \in \mathcal{I}_i)$. $F_i$ represents the first appearance frame of object $o_i$. Then we sort the first appearance frames of all objects to determine the order of appearance:

$$\mathcal{S} = \text{sort}(\{(o_1, F_1), (o_2, F_2), (o_3, F_3)...(o_N, F_N)\}) \tag{10}$$

where $\mathcal{S}$ is the ordered sequence of objects based on their first appearance. For sentence QA type, the ordered sequence and corresponding frame indices are embedded into a sentence. For fill-in-the-blank QA type, the question is to instruct the user to input the ordered sequence as a comma-separated list.

**Obj count** The object counting task estimates the number of object instances for each label category in the scene and generates structured question-answer pairs to facilitate the numerical reasoning ability of LLMs.

For each label $l \in \mathcal{L}$, the total number of object instances is computed as: $N(l) = |\mathcal{L}(l)|$, where $N(l)$ represents the number of object instances associated with label $l$. We will exclude labels with fewer than two object instances.

**Room size** This task is designed for estimating the size of a room and generating corresponding question-answer pairs to facilitate the spatial reasoning ability of LLMs.

Given the room scene with 3D mesh $M = (\mathcal{V}, \mathcal{F})$, where $\mathcal{V}$ is the set of vertices and $\mathcal{F}$ is the set of faces, we first downsample the points by quantizing them into a grid with a specified voxel size $\delta = 0.1$. The quantized points are computed as: $\mathcal{Q} = \lfloor \mathcal{V}/\delta \rfloor$. We then retain only the unique voxels and obtain voxel centers: $\mathcal{P}_d = (\text{unique}(\mathcal{Q}) + 0.5) \cdot \delta$. If the downsampled set contains fewer than 100 points, we revert to the original point cloud. To estimate the room area, we construct a concave hull using the $\alpha$-shape algorithm:

$$\mathcal{H} = \text{AlphaShape}(\mathcal{P}_d[:, 0 : 2], \alpha) \tag{11}$$

where $\alpha = 0.1$ controls the concavity of the shape. The final room area is calculated as:

$$A = \sum_{h \in \mathcal{H}} \text{area}(h) \tag{12}$$

To ensure valid QA generation, If the room area is below the threshold $A_{th} = 5$, no QA pairs will be generated.

**Navigation** We construct visual navigation data based on Matterport3D [61] and Room Across Room (RxR) [62] Dataset. For the navigation instructions and image sequences in RxR, we take the image sequence as input and construct question-answer pairs. We expect the LLM model to complete the absent key action information in the instructions, such as left turn, straight ahead, on the right side, and other keywords.

## 7.6 More visualization

We visualize the detailed QA of different tasks from our proposed *SPAR-7M* in Tab. 12 - 17.

**Table 12: Detailed QA of the Depth Prediction Object-Camera Multi-view Task**

| Task | Question | Answer |
|---|---|---|
| Depth-OC-MV (fill) | The table (**red point**) is located at a depth of 1.5 meters. Estimate the depth of the food container (**blue point**). Calculate or judge based on the 3D center points of these objects. Ensure your answer contains only one number.

 | 1.7 |
| Depth-OC-MV (select) | Given the refrigerator (**red point**) is located at a depth of 0.9 meters in the Z-axis of the camera coordinate system, how far in depth is the dish soap bottle (**blue point**) at its center? Calculate or judge based on the 3D center points of these objects. Please select the correct option from the choices provided. **A. 2.2; B. 2.3; C. 2.1; D. 1.4**. Your answer can only include one of options A, B, C or D.

 | D |
| Depth-OC-MV (sentence) | The wardrobe (**red point**) at a depth of 1.0 meters serves as a reference. How deep is the power socket (**blue point**)? Calculate or judge based on the 3D center points of these objects.

 | With a central depth of 1.2 meters, power socket is referenced here. |

**Table 13: Detailed QA of the Distance Inference Object-Object Task**

| Task | Question | Answer |
|---|---|---|
| DistI-OO (fill) | Between cooking pan (**green point**) and plastic bag (**blue point**), which object is positioned closer to coat (**red point**)? Calculate or judge based on the 3D center points of these objects. Submit your response as the name of one object exclusively.  | Cooking pan |
| DistI-OO (choice) | Which object lies at a closer distance from backpack (**red point**): duffel bag (**green point**) or light switch (**blue point**)? Calculate or judge based on the 3D center points of these objects. Pick the appropriate answer from the options given. **A. duffel bag; B. light switch**. Your answer can only include one of options A, B.  | A |
| DistI-OO (sentence) | Compare the positions of bed (**green point**) and chair (**blue point**). Which is farer to the heater (**red point**)? Calculate or judge based on the 3D center points of these objects.  | The proximity of heater to bed is 2.0 meters, and to chair, it is 0.6 meters. Hence, the bed is farer to heater. |

**Table 14: Detailed QA of the Object Spatial Relation Object-Camera Multi-view Task**

| Task | Question | Answer |
|------|----------|--------|
| ObjRel-OC-MV (choice) | What is the direction of object chair (**bbox**) relative to the observer's primary angle? Calculate or judge based on the 3D center points of these objects. We use the first image to reflect the main perspective, which aligns with the observer's viewpoint. The options describe the spatial relationship between object and observer in terms of left-right (left, right, or empty if indistinguishable), above-below (above, below, or empty if indistinguishable), and front-behind (front, behind, or empty if indistinguishable). Select the correct response from the given choices. **A. left, above, front; B. left, above, behind; C. right, below, front; D. right, above, front**. Your answer can only include one of options A, B, C or D. 

  | C |
| ObjRel-OC-MV (sentence) | Describe the spatial orientation of object bag (**bbox**) relative to the observer. Calculate or judge based on the 3D center points of these objects. The first image is positioned to serve as the main viewpoint for the observer. 

  | Relative to the observer's placement, the bag (**red bbox**) appears to the right below. It seems to the front. |

**Table 15: Detailed QA of the Object Spatial Imagination Object-Camera Multi-view Task**

| Task | Question | Answer |
|------|----------|--------|
| SpImag-OC-MV (choice) | How does the positional relationship of refrigerator (**red bbox**) to the observer evolve once the observer shifts to the 3D center of map poster (**green bbox**) and faces trash can (**blue bbox**)? Calculate or judge based on the 3D center points of these objects. Base your response on the observer's perspective, with the first image defined as the primary view before movement begins. For multiple-choice questions, consider only the state after the observer has moved. The options describe the spatial relationship between object and observer in terms of left-right (left, right, or empty if indistinguishable), above-below (above, below, or empty if indistinguishable), and front-behind (front, behind, or empty if indistinguishable). Select the appropriate option from the given choices. **A. left, , behind; B. right, below, ; C. right, above, ; D. right, below, behind**. Your answer can only include one of options A, B, C or D.  | D |
| SpImag-OC-MV (sentence) | How does the observer's shift to the 3D center of wardrobe (**green bbox**) and orientation toward rug (**blue bbox**) affect the positioning of bread packet (**red bbox**)? Calculate or judge based on the 3D center points of these objects. Frame your answer with the observer's perspective, assigning the first image as the main view before any motion.  | Initially, the position of bread packet appears to the left below to the observer. It is also to the front. After moving to wardrobe and orienting toward rug, bread packet changes to to the right below. It is now to the behind. |

**Table 16: Detailed QA of the View Change Infer Task**

| Task | Question | Answer |
|------|----------|--------|
| ViewChgI (fill) | If starting from the first image, how would the observer's camera need to move to recreate the second image? Provide the camera movement and rotation in the following format: move (right or left):(meters), move (down or up):(meters), move (forward or back):(meters), rotate (down or up):(degrees), rotate (right or left):(degrees) - The first three values are in meters. - The last two values are in degrees. - Use commas to separate each parameter. - Do not include any additional text. Example: move left: 2.6, move down: 0.1, move forward: 0.2, rotate up: 10, rotate left: 0.

 | move right: 1.2, move up: 0.4, move forward: 1.4, rotate up: 5, rotate left: 90 |
| ViewChgI (sentence) | What changes in position or angle would the camera need to make to transition from the first image to the second?

 | In the initial frame, suitcase is located in the bottom-right and moves left and up to the bottom-left in the next image. Observing these changes, it appears the camera movement is as follows: To realign the second image: Move 0.6 meters forward. Shift sideways by 0.2 meters right. Rotate down by 10 degrees. Turn right by 10 degrees. |

**Table 17: Detailed QA of the Camera Pose Task**

| Task | Question | Answer |
|------|----------|--------|
| CamPos (fill) | Estimate the image-plane location and depth in meters of the second image's observer as it appears in the first image's coordinate space. Please ensure your answer is limited to a 2D coordinate and a depth, for instance: $(200, 500)$, 1.2.

 | $(113, 182)$, 3.8 |
| CamPos (choice) | Where would the second image's observer be seen in the first image's space? Provide 2D image-plane coordinates and depth in meters. Choose the correct response from the given choices. **A. Image Coor: (514, 95), Depth:4.8 meters; B. Image Coor: (258, 537), Depth: 6.9 meters; C. Image Coor: (108, 214), Depth: 6.3 meters; D. Image Coor: (921, 261), Depth: 4.6 meters**. Your answer can only include one of options A, B, C or D.

 | A |
| ViewChgI (sentence) | How does the camera's movement between the two images affect its position in the first image? Provide $(X, Y)$ image-plane coordinates and depth in meters.

 | Relative to the first image, the second image's observer occupies the position $(650, 457)$, at a depth of 4.4 meters. |

