# OpenReview forum: "From Flatland to Space: Teaching Vision-Language Models to Perceive and Reason in 3D"
_NeurIPS.cc/2025/Datasets_and_Benchmarks_Track — NeurIPS 2025 Datasets and Benchmarks Track poster_

### Official Review · Reviewer_nXKZ · 2025-06-15

**Rating:** 5
**Confidence:** 4

**Summary:**

This paper presents SPAR-7M, a large vision-language dataset for spatial reasoning from indoor 3D scenes. To cover a broad spectrum of tasks (e.g., viewpoint inference, spatial relations, depth estimation) for single-view, multi-view, and video cases, the authors contribute a variety of tasks. They also introduce SPAR-Bench, a benchmark with manually verified subsets of questions from a larger dataset. Results show enhanced spatial reasoning benchmarks' performance by training on SPAR-7M.

**Dataset Code Accessibility:**

Yes

**Ethical Considerations:**

No, there are no or only very minor ethics concerns

**Final Justification:**

Good dataset. Keep "Accept".

**Limitations Weaknesses:**

Limits applicability in broader environments; outdoor and diverse scenes would help.

Tasks often use artificial visual cues (colored bbox), potentially reducing realism.

Lacks tasks involving semantic or functional spatial reasoning.

Could clarify quality control processes, especially for GPT-generated data.

**Strengths Contributions:**

Covers diverse spatial tasks (single-view, multi-view, video), surpassing existing datasets.

Clear and efficient QA generation from precise 3D annotations.

Demonstrates clear improvements and generalization on existing benchmarks like CV-Bench and VSI-Bench.

Well-organized paper with informative visuals and justified comparisons to prior work.

---

> ### Author Rebuttal · Authors · 2025-07-31
>
> ## Q1:  Limits applicability in broader environment.
>
> Thank you for the valuable feedback. We fully agree that supporting outdoor and diverse environments is important for broad applicability. In this work, we focus on indoor scenes as a starting point to construct a high-quality, tightly controlled benchmark for spatial reasoning. Indoor settings offer dense annotations, reliable 3D reconstructions, and diverse object layouts, which enable rich and precise supervision for training and evaluation.
>
> That said, our data generation pipeline is not restricted to indoor environments. It is fully generalizable and can be applied to outdoor datasets such as Waymo, nuScenes, or KITTI, where multi-view images and pose/depth information are available. This makes it feasible to extend SPAR to cover outdoor navigation, traffic reasoning, or open-space spatial QA in future work.
>
> We believe our current indoor focus offers a strong foundation, and we will explore outdoor extensions as a natural next step.
>
> ## Q2: Visual cues reduce realism.
>
> Thanks for the comment. We clarify that the colored visual cues (e.g., bounding boxes or points) used in some tasks are not embedded into the source images, but are rendered separately by projecting 3D coordinates onto the image plane using camera poses. These visual cues are fully decoupled from the multi-view images, and their inclusion is configurable per task.
>
> For models or settings where image realism is critical, the same spatial information can be provided in coordinate form (e.g., as input tokens or prompts) without rendering any overlays on the images. Our dataset supports both modes, ensuring compatibility with diverse model architectures.
>
> We follow the rendered-cue setting in our experiments for consistency with prior benchmarks (e.g., CVBench), where visual guidance is also explicitly shown during inference. However, our design remains fully modular and can be adapted for realism-sensitive use cases in future deployments.
>
> ## Q3: Lacks tasks involving semantic or functional spatial reasoning.
>
> Thank you for this valuable suggestion. We agree that **semantic and functional spatial reasoning**—such as understanding object affordances or scene functionality—is an important aspect of embodied intelligence.
>
> While the current version of SPAR primarily focuses on **geometric and spatial layout reasoning**, we recognize the value of expanding into **higher-level spatial tasks** that involve semantics, intent, and function.
>
> We will further extent and enhance SPAR’s utility as a **comprehensive benchmark** for evaluating spatial understanding across multiple levels of abstraction.
>
> ## Q4: Clarify quality control processes
>
> Thank you. We clarify that GPT was only used to generate initial question–answer templates, rather the final QA data. For each task type, we prompted GPT to generate approximately 50 question templates and 50 answer templates. These templates were then manually reviewed, filtered, and corrected by researchers to remove any grammatical issues, ambiguity, or ill-formed logic. During dataset construction, a question and an answer template were independently sampled and filled with ground-truth values extracted from the 3D scene to create each final QA instance.
>
> The final QA pairs were constructed automatically by filling these vetted templates with ground-truth values from real 3D scenes (e.g., coordinates, object categories, camera poses). As a result, the QA content is grounded in factual scene-level data, rather the hallucinated generations.
>
> In addition, we conducted a rigorous human verification process on the benchmark split. Seven annotators (PhD/master students) spent over 140 hours reviewing each task type, checking for unclear references, incorrect logic, and label consistency. Each sample was reviewed by at least two annotators, and disagreements were resolved through consensus meetings.
>
> Together, these steps ensure that the final QA data is highly accurate, unambiguous, and reliable, despite originating from a template-based generation pipeline.
>
> While GPT helped seed diverse templates, the final dataset is entirely grounded in scene facts and has been rigorously human-verified.

---

### Official Review · Reviewer_1WeU · 2025-07-01

**Rating:** 5
**Confidence:** 4

**Summary:**

This paper aims to unlock the potential of VLMs for 3D understanding in a implicit manner, without requiring explicit 3D representations. Leveraging ScanNet, ScanNet++, and structured3D, the authors construct a large-scale dataset SPAR-7M. It consists of 4000+ indoor 3D scenes, covering 33 task types with over 7 million QA pairs. To evaluate the performance of models across a wide range of spatial tasks, this paper presents SPAR-Bench. Besides, it takes approximately 140 human hours to evaluate the quality of the benchmark. The evaluation process includes multiple VLM models, including GPT, Cluade, Qwen, InternVL, LLaVA, etc.

**Dataset Code Accessibility:**

Yes

**Dataset Code Comments:**

The guideline for the code and dataset are clear and easy to follow.

**Ethical Considerations:**

No, there are no or only very minor ethics concerns

**Final Justification:**

Thanks for the authors' patient reply. My questions have been solved and I'm inclined to accept this paper.

**Limitations Weaknesses:**

1. The authors mentioned that the data generation pipeline is novel. However, ScanNet, ScanNet++, and Structured3D are widely used datasets for 3D understanding. It would be better to clearly demonstrate the difference between the proposed data generation pipeline with existing benchmarks

Overall, this paper is good, with enough contribution, clear writtiing and comprehensive experimental results. I'm inclined to accept this paper.

**Strengths Contributions:**

1. This paper presents a large-scale dataset, with numerous human efforts.
2. The writting is easy to follow.
3. The dataset is already released on the huggingface, with detailed guideline to use this dataset.
4. The dataset covers a wide range of tasks and scenes.
5. The experimental results are comprehensive.
6. Investigating the potential of VLMs for 3D understanding is an interesting problem.

---

> ### Author Rebuttal · Authors · 2025-07-31
>
> ## Q1: Novelty of data generation pipeline.
>
> We appreciate the reviewer’s insightful question. While existing 3D datasets such as ScanNet, ScanNet++, and Structured3D are widely adopted for scene-level understanding, our work proposes a dedicated QA-oriented data generation pipeline, which differs significantly in both structure and objective. Below, we provide a detailed comparison between our pipeline and several recent multi-view 3D QA or annotation benchmarks, including MSR3D[6], MMScan[7], SceneVerse[8], and RoboSpatial[9].
>
> ### 1. Scene Pre-processing and Structured Scene Database
>
> Our pipeline begins from raw video sequences and constructs a multi-view structured scene database for each environment. We perform keyframe selection to ensure spatial diversity while reducing redundancy, and instead of relying on SfM or depth networks for 3D reconstruction, we directly use ground-truth camera poses and object bounding boxes from the source dataset. Each scene is organized into a database containing synchronized multi-view images, aligned camera intrinsics and extrinsics, and per-frame 3D object metadata. This structured design allows efficient sampling and retrieval of object-centric visual observations across views, and serves as the foundation for generating spatial reasoning tasks involving multiple objects, viewpoints, and coordinate frames.
>
> In contrast, MSR3D focuses on generating situated scene graphs but lacks frame-level alignment across views. MMScan and SceneVerse provide annotations on cropped or segmented regions but do not build a unified object-image-pose database. RoboSpatial leverages existing point cloud annotations, but its focus lies in procedural spatial QA without multi-view image structuring.
>
> ### 2. Configurable Multi-task QA Generation
>
> Our pipeline supports 33 diverse and modular QA task types, spanning a broad spectrum of spatial reasoning levels—from low-level metric estimations (e.g., depth, distance) to high-level multi-view reasoning (e.g., object permanence across views, perspective transformation, 3D layout reconstruction).
>
> These tasks involve varied spatial relations between objects, camera, and scene structure, and are supported by a configurable pipeline that controls task type, object-view selection, reasoning mode (single/multi-view), and answer format. This modular design allows for easy extension to new spatial skills, enabling future addition of functional or semantic reasoning without fundamental changes to the pipeline.
>
> Compared to prior works:
>
> - MMScan and SceneVerse provide a relatively rich set of question types but rely on fixed question templates and lack fine-grained control over view sampling or reasoning granularity.
> - MSR3D also supports question answering but focuses on scene graph-based queries grounded in semantic object relationships, and does not target view-based geometric reasoning.
> - RoboSpatial includes procedural tasks involving spatial transformations and language instructions, but supports fewer task types and lacks multi-view context modeling.
>
> In contrast, our framework is explicitly designed to handle multi-view QA tasks, supports fine-grained spatial relations across views, and provides a scalable task registry that decouples the logic of task design from scene construction. This flexibility is crucial for both benchmarking and training robust 3D vision-language models with broad spatial competence.
>
> We will clarify these points in the paper and provide additional documentation of our task configuration schema to highlight the extensibility of our approach.
>
> ### 3. Template-based but Scene-grounded QA Generation
>
> For each task type, we construct a curated set of approximately 50 question templates and 50 answer templates. These are manually reviewed to ensure linguistic quality, logical clarity, and coverage across scene contexts. During data generation, we randomly pair a question and an answer template, and fill them with ground-truth scene values—such as object names, positions, depths, orientations, and distances—ensuring each QA pair is both factually accurate and linguistically diverse.
>
> This design separates the semantic structure of the question from the scene content, enabling precise control over the reasoning logic while maintaining diversity in phrasing and avoiding hallucination.
>
> In comparison:
>
> - SceneVerse and MMScan also adopt template-based strategies for captioning and QA, but primarily operate over coarse object annotations and lack direct supervision from geometric ground truth (e.g., depth or pose). Their templates are generally more descriptive and less tightly coupled to the underlying 3D scene structure.
> - MSR3D does not support any form of templated QA generation, instead relying on manually annotated question-answer pairs.
>
> In contrast, our pipeline offers both structured reasoning logic and linguistic variation, while being grounded on precise 3D spatial annotations, allowing us to scale the generation of high-quality QA pairs across diverse spatial tasks.
>
> ### Summary
>
> Overall, our pipeline introduces a task-centric, modular, and numerically grounded data generation framework tailored for 3D vision-language tasks. Compared with existing benchmarks, our contributions lie in three main aspects:
>
> - **Structured Multi-view Database**: We construct a unified and queryable database that aligns multi-view images, camera poses, and per-frame object metadata, facilitating object-centric cross-view retrieval.
> - **Flexible Task Configuration**: We support a configurable QA generation system covering a broad range of spatial reasoning types—including metric estimation, relative positioning, viewpoint grounding, and multi-object comparisons.
> - **Grounded and Scalable QA Synthesis**: We enable large-scale QA generation through reusable templates injected with precise scene-derived values, supporting both compositionality and factual correctness.
>
> [6] Multi-modal Situated Reasoning in 3D Scenes, NeurIPS 2024
>
> [7] MMScan: A Multi-Modal 3D Scene Dataset with Hierarchical Grounded Language Annotations, NeurIPS 2024
>
> [8] SceneVerse: Scaling 3D Vision-Language Learning for Grounded Scene Understanding, ECCV 2024
>
> [9] RoboSpatial: Teaching Spatial Understanding to 2D and 3D Vision-Language Models for Robotics, CVPR 2025

---

> > ### Comment · Reviewer_1WeU · 2025-08-09
> > **Reply to the authors from reviewer 1WeU**
> >
> > Thanks for the authors' patient reply. My questions have been solved and I'm inclined to accept this paper.

---

### Official Review · Reviewer_rYzD · 2025-07-01

**Rating:** 4
**Confidence:** 4

**Summary:**

The paper presents SPAR-7M, a vision-language dataset comprising 7 million QA pairs derived from over 4,000 RGB-D indoor scenes. It introduces a 3D-driven generation pipeline that transforms 3D meshes and camera metadata into 33 spatial reasoning tasks across single-view, multi-view, and video settings. Additionally, the authors construct SPAR-Bench, a benchmark consisting of 7,207 manually validated questions spanning 20 representative spatial tasks to evaluate existing methods. Experimental results demonstrate that pretraining on SPAR-7M enhances model performance on both 2D spatial benchmarks and 3D task-specific datasets.

**Additional Feedback:**

(1) The models benchmarked on SPAR-Bench (Full) in Table 4 are relatively small size. If computational resources allow, we encourage the authors to include results using larger VLMs (13B–72B) for a more comprehensive evaluation. It is interesting to see whether SPAR-mix remain beneficial for large-scale VLMs.

(2) In Table 5, what are the models' EM scores on ScanQA?

(3) The authors are encouraged to include more recent literature on multi-view image and 3D understanding benchmarks in the related work or experiment sections [1, 2, 3].

[1] MMSI-Bench: A Benchmark for Multi-Image Spatial Intelligence, Arxiv

[2] Hypo3D: Exploring Hypothetical Reasoning in 3D, ICML 2025

[3] MuirBench: A Comprehensive Benchmark for Robust Multi-image Understanding, ICLR 2025

**Dataset Code Accessibility:**

Yes

**Dataset Code Comments:**

Both SPAR-7M and SPAR-Bench data are open-sourced on Hugging Face. Detailed descriptions of the spatial task pipelines, QA templates, and GPT prompting procedures are available in the supplementary material.

**Ethical Comments:**

The authors have discussed the societal impacts of this paper in the introduction and method sections, indicating no significant ethical concerns.

**Ethical Considerations:**

No, there are no or only very minor ethics concerns

**Final Justification:**

This paper proposes a novel task in 3D/multi-view scene understanding and provide comprehensive experiments and dataset statistics. The author addresses most of my concern in the rebuttal stage. Hence, I recommend accepting this paper.

**Limitations Weaknesses:**

(1) The claim that incorporating 3D representations into VLMs is unnecessary is not entirely convincing. 3D point clouds offer richer and more complete geometric information, and analyzing an entire scene point cloud requires only a single forward pass, whereas multi-view images require multiple passes, increasing computational cost.

(2) SPAR decomposes scenes into multiple “small” scenes using multi-view/single-view images. Why not also provide the corresponding truncated 3D point clouds? Given the camera parameters and images, it should be feasible to include the point cloud regions covered by these views, further enhancing SPAR’s utility for 3D VLM pretraining.

(3) Human accuracy on SPAR-Bench is only ~66%, much lower than the >90% seen on SQA3D and similar benchmarks. Please clarify why SPAR-Bench is more challenging for humans.

**Strengths Contributions:**

(1) SPAR is a large-scale QA dataset for 3D scene understanding, covering a wide range of spatial perception and reasoning tasks. While existing datasets focus mainly on 3D reasoning, SPAR provides extensive data on 3D perception, which is beneficial to the community.

(2)  Performance gains observed on 2D and 3D understanding benchmarks after post-training on the SPAR datasets shows the value of the dataset.

---

> ### Author Rebuttal · Authors · 2025-07-31
>
> ## Q1: 3D representation input is necessary
> Thank you. We agree with both key observations:
>
> - **Yes, 3D point clouds inherently carry more complete and explicit geometric information**, as they directly encode scene structure in metric space;
> - **And yes, a full scene point cloud can be processed in a single forward pass**, whereas multi-view 2D inputs typically require multiple encoder passes.
>
> However, the **practical efficiency and effectiveness** of these approaches depends on more than forward-pass count alone.
>
> First, **multi-view 2D inputs avoid costly 3D preprocessing** steps such as depth map generation, point cloud fusion, voxelization, and sparse-to-dense conversion—all of which are computationally intensive and domain-specific. In practice, **the total runtime of 3D pipelines is often dominated by data preparation**, rather the encoder forward pass itself.
>
> In our experiments (see Reviewer v4TG Q4), our preprocessing takes **0.14s**, compared to **900s** for 3D-LLM and **4.8s** for LEO—despite our multi-pass encoding.
>
> Second, we choose 2D views not to deny 3D’s benefits, but to **leverage strong 2D VLM backbones**, pretrained on large-scale image-text corpora. These offer robust semantic priors and spatial cues, while **3D models lack such scalable pretraining**, limiting generalization.
>
> Finally, point cloud quality is often unreliable across domains, especially outdoors or in low-texture scenes. **Multi-view images are easier to obtain, simulate, and standardize**, making them more robust for large-scale use.
>
> In summary, while 3D inputs offer high geometric fidelity, multi-view 2D provides a more scalable, efficient, and transferable alternative under current VLM architectures—striking a practical trade-off between geometric completeness and real-world usability.
>
> ## Q2: Truncated 3D point clouds
>
> Thank you for the thoughtful suggestion. We agree that including truncated 3D point clouds for each multi-view scene could further benefit 3D VLM pretraining.
>
> While SPAR does not directly include precomputed point clouds, we **do provide all necessary ingredients to reconstruct them**: each scene includes **multi-view RGB images, ground-truth depth maps, and calibrated camera poses**. These enable users to easily generate **region-specific 3D point clouds via standard depth back-projection**, if desired. This design keeps the dataset lightweight while offering flexibility to support 3D extensions.
>
> In this paper, our goal is to **demonstrate that multi-view (or even single-view) 2D images, when paired with spatial prompts,** offer **enough spatial information to enable plausible 3D reasoning without explicit 3D input**. We show that pretrained VLMs can leverage this structure to perform well across diverse spatial tasks—without relying on explicit 3D inputs.
>
> That said, we view SPAR as a **modular framework**. Providing preprocessed 3D representations is a natural extension, and we plan to include them in future releases to further broaden SPAR's utility for 3D-aware pretraining and evaluation.
>
> ## Q3: Human accuracy
>
> Thank you for the insightful observation. We agree that the ~66% human accuracy on SPAR-Bench appears lower than benchmarks such as SQA3D, and we appreciate the opportunity to clarify the source of this discrepancy.
>
> SPAR-Bench is designed to assess a wider and more demanding spectrum of spatial reasoning, from low-level metric estimations (e.g., depth, distance) to higher-level tasks like multi-view integration and spatial imagination—resulting in greater cognitive load for human annotators.
>
> We identify two primary factors contributing to lower human performance:
>
> 1. **Metric Depth and Distance Estimation**: Humans often struggle to judge absolute or relative depth from monocular 2D images, especially without stereoscopic or motion cues. SPAR-Bench includes two notably difficult tasks: (1) **estimating Euclidean distance between object centers across views**, which requires mentally aligning positions from different perspectives, and (2) **estimating depth differences between objects from a single view**—not just which is closer, but by how much. Both are error-prone, particularly without strong perspective cues or when objects appear similarly sized. These tasks significantly lowered human accuracy. Additionally, our annotators were not professionally trained and relied solely on visual inspection, simulating average user-level reasoning.
> 2. **Multi-view Integration and 3D Imagination**: Many tasks require integrating **information across multiple viewpoints** and mentally reconstructing a coherent 3D layout. This form of **perspective-taking** and **scene imagination** is known to be cognitively demanding, especially in the absence of interactive feedback or physical reference. In some cases, **minor viewpoint shifts** or **visual ambiguities** (e.g., partial occlusions, inconsistent lighting) can further confuse human annotators.
>
> Overall, we believe that the lower human accuracy reflects the **greater realism and difficulty of SPAR-Bench**, which better captures the challenges VLMs may face in practical embodied settings.
>
> ## Q4: Larger VLMs on SPAR-Bench (Full)
>
> Table1. Performance comparison on SPAR-Bench (Full) across large-scale vision-language models
>
> | Full | Ave. | SPAR-Low | SPAR-Med | SPAR-High |
> | --- | --- | --- | --- | --- |
> | GPT-4.1 | 41.60 | 41.95 | 44.02 | 42.93 |
> | GPT-4o | 38.11 | 36.88 | 26.49 | 43.80 |
> | Doubao-1.5-vision-pro | 41.74 | 33.24 | 47.36 | 49.42 |
> | Qwen2.5-VL-72B-Instruct | 37.01 | 29.94 | 44.61 | 43.80 |
> | Qwen2.5-VL-32B-Instruct | 33.09 | 27.09 | 33.92 | 40.44 |
> | InternVL2.5-38B | 33.83 | 25.99 | 30.88 | 44.13 |
> | InternVL2.5-26B | 34.11 | 24.18 | 50.11 | 42.40 |
> | LLaVA-v1.5-13B | 28.62 | 25.92 | 31.03 | 32.33 |
> | SPAR-mix-8B | 63.25 | 65.53 | 63.01 | 61.29 |
>
> Thank you for the helpful suggestion. We fully agree that evaluating larger VLMs would further enrich the benchmark analysis. Although current computational constraints prevent us from fine-tuning SPAR-mix on 13B–72B models, we do include several strong large-scale baselines in Table 1—such as GPT-4.1, GPT-4o, Qwen2.5-VL-72B, and InternVL2.5-38B.
>
> Interestingly, despite their scale, none of these models surpasses the SPAR-mix fine-tuned model or human-level performance on SPAR-Bench.
>
> These results suggest that strong spatial reasoning does not emerge solely from scaling, and that targeted spatial supervision, such as that provided by SPAR-mix, remains necessary even for large VLMs. We believe this highlights the practical value of our dataset and training strategy, and supports its utility as a diagnostic tool for improving next-generation models.
>
> We plan to extend our experiments to larger model variants as resources permit in future work.
>
> ## Q5: EM scores on ScanQA
>
> Table 2: Comparison of SQA3D and ScanQA performance. “3D” indicates whether the model is infused with 3D information.
>
> | Methods         | 3D  | EM@1 (SQA3D) | BLEU-4 (ScanQA) | CiDEr (ScanQA) | EM (ScanQA) |
> |----------------|-----|--------------|------------------|----------------|--------------|
> | 3D-LLM         | ✓   | -            | 12.0             | 69.4           | 20.4         |
> | Chat-3D v2     | ✓   | 54.7         | 14.0             | 87.6           | -            |
> | LEO            | ✓   | 50.0         | 13.2             | 101.4          | 21.5         |
> | LL3DA          | ✓   | -            | 13.5             | 76.8           | -            |
> | Scene-LLM      | ✓   | 54.2         | 12.0             | 80.0           | 27.2         |
> | LLaVA-3D       | ✓   | 55.6         | 14.5             | 91.7           | 27.0         |
> | Video-3D LLM   | ✓   | 58.6     | 16.2         | 102.1      | 30.1         |
> | *SPAR-mix*     | ✗   | 58.1         | 15.3             | 90.7           | 27.7         |
>
> Thank you for pointing this out. We acknowledge that the EM score of SPAR-mix on ScanQA (27.7) is slightly lower than that of some 3D-enhanced models such as Video-3D LLM (30.1). However, we would like to highlight a few key points:
>
> SPAR-mix does not rely on any explicit 3D input or supervision during training or inference, yet still achieves competitive or superior performance on semantic-oriented metrics like BLEU-4 (15.3) and CiDEr (90.7). These metrics better reflect spatial reasoning and generation quality than surface-form exact match.
>
> The slightly lower EM score is largely due to surface-level phrasing differences. As a generative model, SPAR-mix may produce correct but lexically different answers (e.g., synonyms, rephrasings), which EM does not account for. This is a known limitation of EM in open-ended QA settings.
>
> We believe this EM gap can be further narrowed by incorporating answer normalization, span-level supervision, or light postprocessing, which are orthogonal to our main contributions and left as future extensions.
>
> In summary, SPAR-mix offers strong performance with 2D-only inputs, and its results underscore the viability of efficient, geometry-free models for spatial question answering.
>
> ## Q6: Include more recent literature.
>
> Thank you for the helpful suggestion. We appreciate the reviewer’s pointer to recent benchmarks on multi-view image and 3D understanding [3, 4, 5]. We will cite and discuss your recommended literature on multi-view image and 3D understanding benchmarks in the revised version of the paper.
>
> [3] MMSI-Bench: A Benchmark for Multi-Image Spatial Intelligence, Arxiv
>
> [4] Hypo3D: Exploring Hypothetical Reasoning in 3D, ICML 2025
>
> [5] MuirBench: A Comprehensive Benchmark for Robust Multi-image Understanding, ICLR 2025

---

### Official Review · Reviewer_v4TG · 2025-07-02

**Rating:** 5
**Confidence:** 4

**Summary:**

This paper, "From Flatland to Space: Teaching Vision-Language Models to Perceive and Reason in 3D," introduces an approach to equip Vision-Language Models (VLMs) with 3D spatial understanding without requiring direct 3D data as input. The core method involves an automated pipeline that uses 3D ground-truth from existing scenes to generate SPAR-7M, a large-scale dataset of 2D images paired with spatial questions and answers. By training on this dataset, the model learns to perform 33 distinct spatial tasks, from basic perception to complex reasoning, using only 2D images during inference. The paper also presents SPAR-Bench, a comprehensive benchmark designed for the systematic evaluation of these spatial abilities.

**Additional Feedback:**

* A quantitative analysis of the inference efficiency would be a powerful addition. Since the proposed approach avoids heavy 3D encoders, demonstrating concrete improvements in latency and memory usage would strongly support its adoption in resource-constrained applications like robotics or AR.
* It would be fascinating to investigate the nature of the "implicit 3D representation" the model learns. Perhaps the model's internal features could be probed to generate outputs like depth maps or perform novel view synthesis. Such an analysis would provide deeper insights into how the model builds its spatial understanding, moving beyond its performance on downstream tasks.

**Dataset Code Accessibility:**

Yes

**Dataset Code Comments:**

The dataset and benchmark is accessible

**Ethical Considerations:**

No, there are no or only very minor ethics concerns

**Final Justification:**

Author's rebuttal resolves my concerns, I'd like keep my rating as Accept

**Limitations Weaknesses:**

* Training Data Dependency: The method relies on high-quality 3D data to generate its training set. While the model uses 2D images for inference, this dependency on 3D ground-truth could limit its scalability to new domains where such data is unavailable.
* Focus on Static Tasks: The paper primarily addresses static question-answering. This scope, while broad, does not fully cover the dynamic and interactive reasoning required for embodied AI applications like robotics.
* Theoretical Performance Limits: From an information theory standpoint, relying on 2D projections may have a lower performance ceiling than methods that process full 3D data, especially for tasks that demand high geometric precision or must handle significant object occlusion.

**Strengths Contributions:**

* Practical Paradigm: The approach is innovative and practical, demonstrating that VLMs can learn 3D reasoning without specialized 3D encoders. This allows for the direct use of established 2D VLM architectures and their extensive pre-trained weights.
* Valuable Resources: The SPAR-7M dataset and SPAR-Bench benchmark are significant contributions. They are notable for their scale, task diversity, structured difficulty levels, and support for multiple input formats (single-view, multi-view, and video).
* Thorough Validation: The work is supported by solid experimental validation. The method's effectiveness is clearly demonstrated through comparisons with a range of contemporary models on multiple benchmarks.

---

> ### Author Rebuttal · Authors · 2025-07-31
>
> ## Q1: Training Data Dependency
>
> Thank you. We agree that high-quality 3D ground-truth data may not always be available across domains, and we appreciate the opportunity to clarify the scalability of our method.
>
> When ground-truth point clouds are unavailable, we can reconstruct scene geometry from monocular video using standard methods like SfM, NeRF, or 3D Gaussian Splatting—all of which require only RGB input and are applicable to real-world settings.
>
> We acknowledge, however, that such monocular reconstructions typically lack absolute scale. As a result, tasks involving metric predictions—such as absolute depth or real-world distance estimation—may suffer without additional information.
>
> To address this limitation, we propose the following solutions:
>
> - For relational spatial tasks (e.g., reasoning about which object is closer, relative directions, or scene-level geometry), relative scale is sufficient, and our full pipeline can be applied directly.
> - For metric tasks, a lightweight scale recovery step (e.g., known object sizes, stereo cues, or device AR frameworks) can be used to align the reconstructed scene to metric scale with minimal additional cost.
> - Alternatively, we can reformulate metric tasks (e.g., using normalized or ordinal targets) to remain robust even under scale ambiguity.
>
> Overall, our approach is designed to be flexible: while accurate metric 3D data enhances supervision quality, it is not strictly required for deploying our method in new domains. We will revise the paper to clarify these points and to explicitly state the assumptions and options for scale handling when ground-truth 3D data is unavailable.
>
> ## Q2: Focus on Static Tasks
>
> Thank you for the insightful comment. While our current work focuses on static spatial reasoning, we view it as a critical foundation for embodied intelligence. Understanding 3D structures, relative positions, and affordances in static scenes is essential for downstream tasks such as navigation, manipulation, and spatial dialogue.
>
> SPAR-7M includes 33 diverse task types across over 4,000 scenes, targeting core spatial competencies through multi-view, multi-object QA. These static settings offer a scalable, controlled testbed for developing general-purpose spatial priors.
>
> In future work, we are actively extending our framework to dynamic and egocentric scenarios, including video-based reasoning and multi-turn spatial dialogue, and appreciate the opportunity to clarify this direction.
>
> ## Q3: Theoretical Performance Limits
>
> Thank you for the thoughtful comment. While we agree that full 3D input theoretically offers richer geometric cues, our choice of 2D projections is driven by practical trade-offs and scalability:
>
> 1. **Model maturity and scalability**: 3D transformers remain less mature, more resource-intensive, and lack large-scale pretraining data compared to 2D models.
> 2. **Leverage of 2D priors**: Using pretrained 2D encoders enables us to inherit strong semantic priors from large-scale image-text data, which are critical for language-grounded spatial reasoning.
> 3. **Multi-view compensation**: By aggregating multiple views, models can recover spatial cues and mitigate occlusion without explicit 3D input—yielding strong empirical performance (see Tables 5 & 6).
>
> While 3D representations offer theoretical advantages, they require richer supervision and alignment with language, which remains challenging at scale. Our approach offers a practical and effective alternative, and we plan to explore hybrid 2D+3D models in future work.
>
> ## Q4: A quantitative analysis of the inference efficiency
>
> Thank you for the valuable suggestion. We agree that demonstrating concrete inference efficiency is important for real-world applications such as robotics or AR. To this end, we conducted a quantitative latency and memory analysis of our model compared to several 3D-VLM baselines, including LLaVA-3D, LL3DA, and LEO.
>
> We report feature extraction latency (from raw modality to encoder output), inference latency (encoder + LLM), peak GPU memory usage, and ScanQA CiDEr performance. All models were evaluated on an A6000 GPU using bf16 and FlashAttention.
>
> Table 1: Inference efficiency comparison across VLMs using 3D vs. 2D inputs.
> | Model             | Input       | Feature Extraction Latency (s) | Inference Latency (ms) | Peak Memory (MB) | CiDEr (ScanQA) |
> |------------------|-------------|-------------------------------|-------------------------|------------------|----------------|
> | 3D-LLM | 3D | 900.0| 197.4 | 16076.4          | 69.4           |
> | LL3DA            | 3D          | 0.3  | 382.0 | 3306.4           | 76.8           |
> | LEO              | 3D          | 4.8 | 505.6  | 15320.0          | 101.4          |
> | LLaVA‑3D         | 3D          | 0.2  | 494.1                   | 15313.5          | 91.7           |
> | Ours 13×448×448  | 2D          | 0.14  | 771.1                   | 19473.7          | 90.7           |
> | Ours 11×448×448  | 2D          | 0.12 | 662.1                   | 18942.8          | 89.6           |
> | Ours 13×224×224  | 2D          | 0.04                    | 245.6                   | 16982.1          | 87.6           |
>
> - **No 3D preprocessing overhead**: Unlike 3D-based models, our pipeline requires no point cloud generation or voxel/mesh conversion. The 0.14s feature latency reflects standard image loading and resizing, making our model highly suitable for deployment pipelines without 3D sensors.
> - **Tradeoff in input token volume**: Our default setting (13×256 tokens) increases sequence length, but both frame count and resolution are configurable. We also report results with 11×448×448 and 13×224×224 inputs, showing only slight performance drops while achieving significantly faster inference.
> - **Memory usage**: Although our model avoids 3D processing overhead, the use of multi-frame 2D input introduces higher peak memory usage than expected, primarily due to the large number of vision tokens passed to the transformer encoder.
>
> Overall, while we do not claim state-of-the-art efficiency in its current form, our framework offers 3D-free inference, avoids geometric preprocessing overhead, and provides a clear path toward further optimization via sequence pruning and temporal sparsity.
>
> [1] LLaVA-3D: A Simple yet Effective Pathway to Empowering LMMs with 3D-awareness, ICCV 2025
>
> ## Q5: Investigating Implicit 3D Representations Learned by the Model
>
> Thank you for this insightful suggestion. We fully agree that probing the model’s implicit 3D representation is essential to understanding how spatial understanding emerges beyond standard QA metrics.
>
> To investigate this, we design a language-guided cognitive map construction task, inspired by [2], where the model is prompted to produce 2D Bird’s-Eye View (BEV) coordinates of multiple objects based on multi-view 2D inputs. The task requires the model to mentally reconstruct a scene-centric world coordinate system, infer the 3D layout, and project each object onto the horizontal plane—purely from image inputs, with no 3D supervision at inference.
>
> Specifically, we construct a val set of 1,000 samples sampled from the ScanNet++ validation split, where each instance contains 3 images and 3-7 annotated objects with known 3D positions.
>
> For each instance, the model is prompted to generate **2D Bird’s-Eye View (BEV) coordinates of objects in an observer-centric coordinate frame**, where the origin is anchored at the main view (first image), the Y-axis aligns with the projected viewing direction, and the X-axis points to the observer’s right on the ground plane. The prompt provides detailed geometric rules in natural language, specifying how to construct the local coordinate frame and how to report coordinates in meters using a fixed structured format
>
> We evaluate performance using **Average Position Error (APE)** between predicted and ground-truth BEV coordinates, reporting object-level results. We compare our SPAR-mix model with Base VLM (InternVL2.5‑8B, a pretrained 2D-only VLM). As shown below:
>
> Table 2. Probing implicit spatial representations via BEV coordinate prediction.
>
> | Model | Obj APE Mean ↓ | Obj APE P50 ↓ | Obj APE P90 ↓ | Format OK |
> | --- | --- | --- | --- | --- |
> | Base VLM | 2.22 | 1.86 | 4.12 | 86.7% |
> | **SPAR-mix** | **1.05** | **0.64** | **2.45** | 100.0% |
>
> SPAR-mix shows significantly **lower spatial error**, indicating stronger spatial priors and internal geometric structuring.
>
> To further understand how distance affects model accuracy, we analyze APE across seven distance bins (based on ground-truth object distance from the main view). SPAR-mix consistently outperforms Base VLM, with increasing margins in the medium-to-far range, where depth and occlusion reasoning become more difficult:
>
> Table3. Object-level BEV prediction accuracy across distance bins.
>
> | Distance Bin (m) | #Objs | APE Mean ↓ (Ours) | APE Mean ↓ (Base VLM) | APE P90 ↓ (Ours) | APE P90 ↓ (Base VLM) |
> | --- | --- | --- | --- | --- | --- |
> | [0, 1) | 990 | **0.70** | 0.74 | 1.61 | **1.03** |
> | [1, 2) | 2635 | **0.73** | 1.40 | **1.77** | 1.88 |
> | [2, 3) | 1619 | **1.04** | 2.39 | **2.25** | 2.86 |
> | [3, 5) | 1225 | **1.46** | 3.66 | **3.26** | 4.50 |
> | [5, 7) | 346 | **2.41** | 5.62 | **4.64** | 6.55 |
> | [7, 10) | 62 | **4.43** | 7.62 | **7.38** | 8.42 |
> | [10, ∞) | 5 | **11.57** | 12.71 | **15.26** | 16.46 |
>
> These results indicate that SPAR-mix learns a scene-consistent, object-level spatial layout, aggregating visual cues across views to infer geometric relationships—even without any explicit 3D input during inference. This probing analysis shows that our model develops internal geometric understanding, offering a foundation for future extensions such as view synthesis, depth prediction, or spatial simulation.
>
> We appreciate this suggestion and will include this analysis in the final version of the paper.
>
> [2] Thinking in Space: How Multimodal Large Language Models See, Remember, and Recall Spaces, CVPR 2025

---

> > ### Comment · Reviewer_v4TG · 2025-08-04
> >
> > Thanks authors for the detailed rebuttal, it resolves all my concerns. I'd keep my rating as Accept

---

### Decision · Program_Chairs · 2025-09-18

**Decision:**

Accept (poster)

**Comment:**

The paper proposed SPAR-7M dataset with SPAR-Bench to facilitate 3D spatial understanding. The key idea is to first extract the 3D information (including the bboxes, camera poses) and the corresponding image key frames (along with the 2D projections). The 3D-related tasks and QA pairs are then constructed through templates and refined by GPT.

The key strength is the diversity of the dataset in the paper, as it spans across 33 different tasks, ranging from low-level perception to high-level reasoning, and spans different input modalities (single-view, multi-view and video). The dataset is then verified through multiple experiments through training 3D grounded spatial reasoning VLMs.

Some weaknesses exist, including the limitation to mostly static scenes and building on top of existing datasets.

After rebuttals, all the reviewers appreciate the contributions from this paper; the AC follows the consensus and recommends accepting the paper.